# Novelty and uncertainty differentially drive exploration across development

Kate Nussenbaum[1†], Rebecca E Martin[1†], Sean Maulhardt[1,2], Yi (Jen) Yang[1,3], Greer Bizzell-Hatcher[1], Naiti S Bhatt[1], Maximilian Koenig[1,4], Gail M Rosenbaum[1,5], John P O'Doherty[6], Jeffrey Cockburn[6], Catherine A Hartley[1]*

[1]New York University, New York, United States; [2]University of Maryland, College Park, United States; [3]Temple University, Philadelphia, United States; [4]Leiden University, Leiden, Netherlands; [5]Geisinger Health System, Danville, United States; [6]Caltech, Pasadena, United States

**Abstract** Across the lifespan, individuals frequently choose between exploiting known rewarding options or exploring unknown alternatives. A large body of work has suggested that children may explore more than adults. However, because novelty and reward uncertainty are often correlated, it is unclear how they differentially influence decision-making across development. Here, children, adolescents, and adults (ages 8–27 years, $N$ = 122) completed an adapted version of a recently developed value-guided decision-making task that decouples novelty and uncertainty. In line with prior studies, we found that exploration decreased with increasing age. Critically, participants of all ages demonstrated a similar bias to select choice options with greater novelty, whereas aversion to reward uncertainty increased into adulthood. Computational modeling of participant choices revealed that whereas adolescents and adults demonstrated attenuated uncertainty aversion for more novel choice options, children's choices were not influenced by reward uncertainty.

*For correspondence:
cate@nyu.edu

†These authors contributed equally to this work

## Editor's evaluation

This is an important study that investigates changes in novelty-seeking and uncertainty-directed exploration from childhood to adulthood. A wide age range of participants was tested using a well-suited task and performance was analyzed using a sophisticated model-based approach. The results provide compelling evidence that age-related changes in decision-making are driven by attenuation in uncertainty aversion in the presence of stable novelty-seeking from childhood to adulthood.

## Introduction

Across the lifespan, exploration increases individuals' knowledge of the world and promotes the discovery of rewarding actions. In some circumstances, exploring new options may yield greater benefits than sticking to known alternatives, whereas in others, 'exploiting' known options may bring about greater rewards. This trade-off is known as the 'explore–exploit' dilemma (*Cohen et al., 2007*; *Sutton et al., 1998*), reflecting the challenge inherent to resolving this tension. In general, the optimal balance between exploration and exploitation may shift across the lifespan. Relative to adults, children tend to know less about the world and have longer temporal horizons over which to exploit newly discovered information (*Gopnik, 2020*; *Gopnik et al., 2017*). Thus, it may be advantageous to explore to a greater extent earlier in life, and gradually shift to a more exploitative decision strategy as experience yields knowledge. Empirical data suggest that individuals at varied developmental stages do indeed tackle explore–exploit problems differently. Children and adolescents tend to explore more than adults (*Christakou et al., 2013*; *Giron et al., 2022*; *Jepma et al., 2020*; *Lloyd et al.,*

*2021*; *Nussenbaum and Hartley, 2019*; *Schulz et al., 2019*), and this increased exploration promotes enhanced learning about the structure of the environment (*Blanco and Sloutsky, 2021*; *Liquin and Gopnik, 2022*; *Sumner et al., 2019*). Despite compelling arguments for why an early bias toward exploration may be advantageous and growing evidence that children *are* in fact more exploratory than adults, the cause of the developmental shift toward exploitation remains unclear.

Prior work has revealed that across development, two features of choice options influence exploration: stimulus *novelty* (*Daffner et al., 1998*; *Gottlieb et al., 2013*; *Henderson and Moore, 1980*; *Jaegle et al., 2019*; *Kakade and Dayan, 2002*; *Wittmann et al., 2008*) and reward *uncertainty* (*Badre et al., 2012*; *Blanco and Sloutsky, 2021*; *Gershman, 2018*; *Somerville et al., 2017*; *Trudel et al., 2021*; *Wang et al., 2021*; *Wilson et al., 2014*). Here, we use *novelty* to refer to the extent to which choice options have been previously encountered and *uncertainty* to refer to the variance in the distributions of rewards that they yield. Disentangling the role of novelty and uncertainty in driving exploratory decision making is challenging because they are often correlated. For example, a new toy has high novelty because it has never been encountered and high reward uncertainty because its entertainment value is unknown. Still, while novel stimuli almost always have high reward uncertainty, in many cases, *familiar* options do as well — when buying a familiar toy as a gift for *someone else*, one may have little knowledge of how much they will like it.

A recent study in adults took advantage of these types of choices, and, by harnessing familiar options with unknown reward probabilities, decoupled the influence of novelty and uncertainty on exploratory decision making in adults (*Cockburn et al., 2022*). Adults were novelty-seeking, preferentially selecting choice options that they had encountered infrequently in the past versus those that were more familiar. However, adults were also uncertainty averse, such that they tended to avoid options with high reward uncertainty. This tension between avoiding uncertain options while pursuing novel alternatives, which are themselves inherently uncertain, suggested *interactive* effects of choice features. Computational modeling further revealed that stimulus novelty diminished the influence of uncertainty on exploratory choice. Thus, these findings suggest that value-guided decision making in adults — and specifically, the balance between exploration and exploitation — may be governed by complex interactions among different features of choice options. To date, however, the influences of novelty and uncertainty have not been disentangled in children and adolescents.

Changes in the influence of these choice features may shift the explore–exploit balance across development. A stronger appetitive influence of stimulus novelty may drive greater exploration earlier in life. Reduced uncertainty aversion, or perhaps even an early preference to explore more uncertain options, may similarly promote heightened exploratory behavior. Novelty and uncertainty may also exert unique, interactive effects for younger individuals. Though prior studies have found effects of novelty (*Henderson and Moore, 1980*; *Mendel, 1965*) and reward uncertainty (*Blanco and Sloutsky, 2021*; *Meder et al., 2021*; *Schulz et al., 2019*) on exploration and choice in early childhood, it is unclear how their relative influence changes from childhood to early adulthood, leaving open the question of *why* children tend to explore more than adults. Further, in most developmental studies, novelty and uncertainty are confounded, making it difficult to tease apart their separate, motivational effects.

Here, using an adapted version of the task introduced in *Cockburn et al., 2022* with a large age-continuous developmental sample, we asked how the influence of novelty and uncertainty on exploratory choice changes from middle childhood to early adulthood. We hypothesized that the developmental shift from more exploratory to more exploitative behavior would be driven by changes in how both novelty and reward uncertainty affect the evaluation of choice options from childhood to adulthood.

## Results

### Approach

Participants ages 8–27 years (*N* = 122; mean age = 17.9 years, standard deviation [SD] age = 5.6 years, 62 females, 59 males, 1 non-binary) completed a child-friendly decision-making task adapted from one used in a prior adult study (*Cockburn et al., 2022*). In the task, participants tried to find gold coins that various creatures had hidden in different territories. The task was divided into 10 blocks of 15 trials. Each block took place within a different territory in which coins were hidden by a distinct

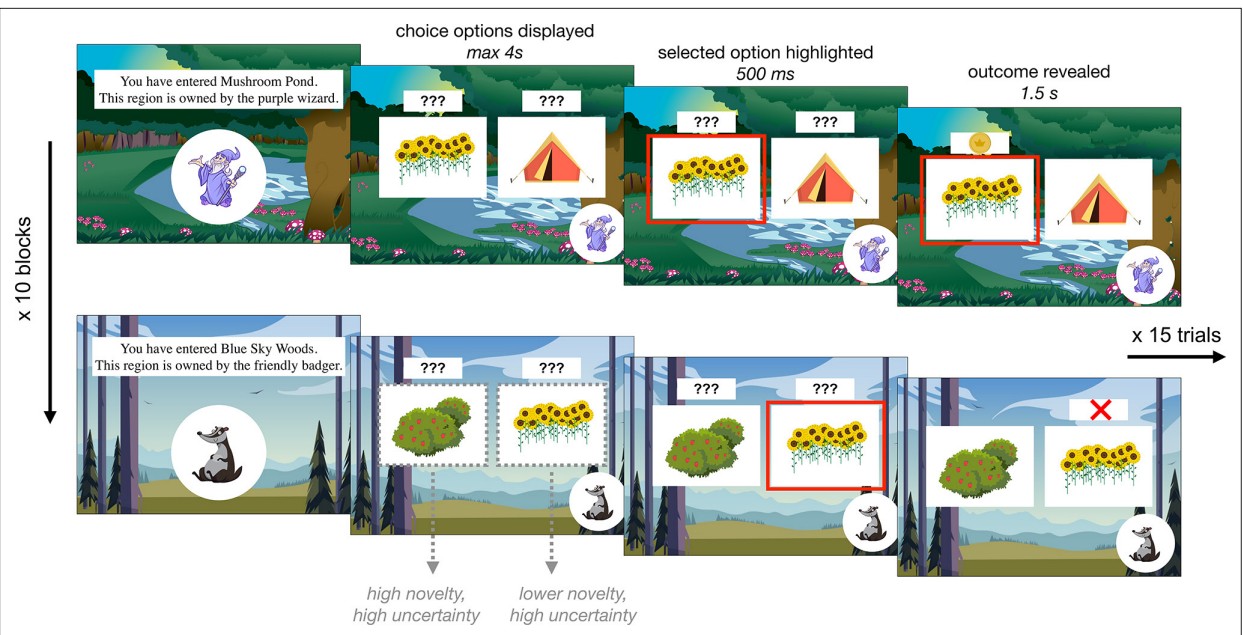

**Figure 1.** Exploration task. Participants completed 10 blocks of 15 choice trials in which they selected between two of three 'hiding spots' to find gold coins. Within each block, two hiding spots had been previously encountered and one was completely novel. Each block took place within a different 'territory' in which a new creature hid coins. Each creature had different preferred hiding spots, such that the reward probabilities associated with each option were reset at the beginning of each block.

creature. Each creature hid their coins among three possible locations, one of which held a coin on either 20 or 30% of trials (on easy and hard blocks, respectively), one of which held a coin on 50% of trials, and one of which held a coin on either 70 or 80% of trials. Participants were not explicitly informed of these probabilities, and had to learn, through trial and error, where each creature was most likely to hide a coin. On every trial, participants viewed two hiding spots and had to select one in which to search for a coin (*Figure 1*). After a brief delay, participants saw the outcome of their choice — either a coin or an X indicating that they had not found a coin. Throughout each block, the background of the screen indicated the territory and a picture in the lower left corner indicated the creature that had hidden coins there.

Importantly, after the first block, each subsequent block contained two hiding spots that participants had encountered in previous blocks and one novel hiding spot they had not seen before. Though participants had already encountered two of the hiding spots within each block, the reward probabilities were re-randomized for every creature. In this way, the task dissociated sensory *novelty* and reward *uncertainty*. At the beginning of every block, the novelty of each hiding spot varied but *all* hiding spots had high reward uncertainty. Participants were explicitly told that the reward probabilities were reset in every block; within the task narrative, this was framed as each creature having different favorite hiding spots in their respective territory (see Appendix 1 for analyses demonstrating that participants of all ages indeed comprehended these instructions and 'reset' the reward probabilities at the beginning of each block). After the exploration task, participants completed a surprise memory test in which they were shown each of the ten creatures, one at a time, and asked to select its favorite hiding spot from an array of five options.

## Exploration task performance

First, we examined whether participants learned to select the better options within each block of the task. On each trial, the optimal choice was defined as the option with the higher reward probability. A mixed-effects logistic regression examining the effects of within-block trial number, age, block difficulty, block number, and their interactions, with random participant intercepts and slopes across trial number, block difficulty, and block number revealed that participants learned to make more optimal choices over the course of each block, odds ratio (OR) = 1.11, 95% confidence interval = [1.07, 1.16], $\chi^2$ (1) = 24.5, $p$ < 0.001. In addition, participants learned faster, OR = 1.05 [1.01, 1.08], $\chi^2$ (1) = 8.2, $p$

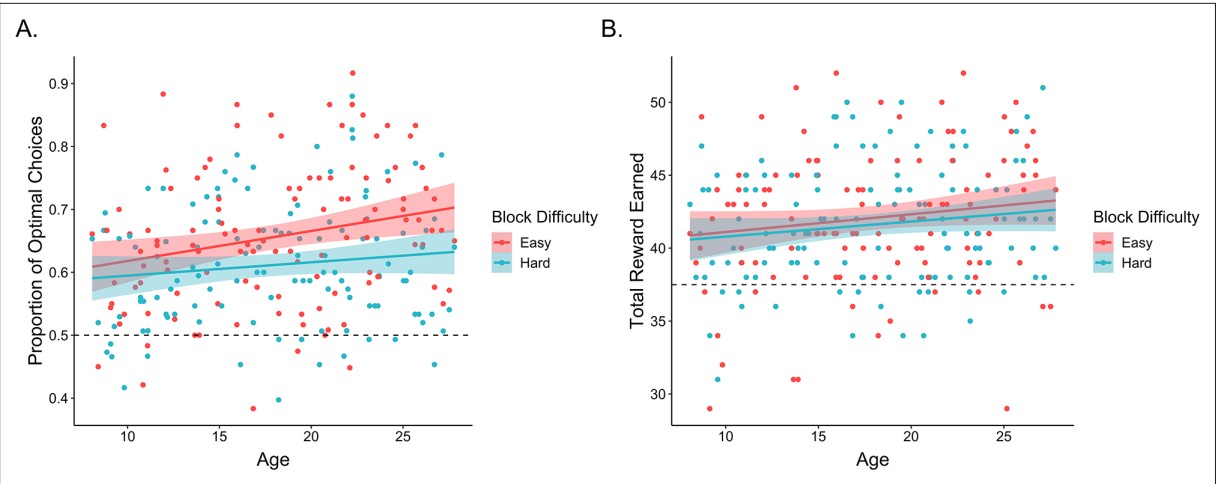

**Figure 2.** Exploration task performance. (**A**) Participants' (n = 122) proportion of optimal choices as a function of age and block difficulty. (**B**) Reward participants earned in easy and hard blocks of the task across age. In both plots, points represent participant averages in each block condition, lines show the best-fitting linear regression modeling the effect of age, and the shaded regions around them represent 95% confidence intervals. The dotted lines indicate chance-level performance.

= 0.004, and made more optimal choices in easy relative to hard blocks, OR = 1.11 [1.07, 1.15], $\chi^2$ (1) = 25.5, $p$ < 0.001 (*Figure 2A*). Performance also improved with increasing age, OR = 1.10 [1.03, 1.18], $\chi^2$ (1) = 7.1, $p$ = 0.008 (*Figure 2A*). While we did not observe a main effect of block number, we did observe a block number × block difficulty interaction, OR = 0.96 [0.93, 0.99], $\chi^2$ (1) = 7.3, $p$ = 0.007, as well as a block difficulty × trial × block number interaction, OR = 0.94 [0.91, 0.97], $p$ < 0.001, such that performance differences between easy and hard blocks were greater earlier in the experiment. No other interactions reached significance ($ps$ > 0.07).

We followed the same analytical approach to examine whether participants earned reward on each trial, though we removed the block difficulty and block number random slopes to allow the model to converge. Here, we similarly observed that performance improved across trials within each block, OR = 1.03 [1.00, 1.07], $\chi^2$ (1) = 3.9, $p$ = 0.049. Further, the rate at which participants learned to make rewarding choices was faster in easier versus harder blocks, OR = 1.03 [1.00, 1.06], $\chi^2$ (1) = 3.9, $p$ = 0.048, though the effect of block difficulty was greater in earlier blocks, OR = 0.97 [0.94, 1.00], $\chi^2$ (1) = 4.0, $p$ = 0.045. No other main effects or interactions reached significance.

Taken together, these findings indicate that participants across our age range learned to select rewarding choice options throughout each block, though the extent to which participants learned to 'exploit' the most rewarding choice options increased across age. Participants also demonstrated above-chance (defined as 0.2) memory for the most rewarding choice within each block (mean = 0.25; standard error [SE] = 0.001; $t$(119) = 4.02, $p$ < 0.001). Memory accuracy did not vary across age, $\beta$ = −0.046, SE = 0.07, $z$ = −0.69, $p$ = 0.49, and did not significantly relate to individual differences in learning (see Appendix 1).

## Age-related change in exploration

Next, we turned to our main questions of interest: whether and how novelty and uncertainty influenced exploration across age. To examine the influence of expected value, uncertainty, and stimulus novelty on choice behavior, we defined and computed these three feature values for each choice option on every trial (*Cockburn et al., 2022*). Expected value was defined as the mean of the beta distribution specified according to the win and loss history of each choice option (hiding spot) within the block: $\frac{\alpha}{\alpha+\beta}$ where $\alpha$ = number of wins +1 and $\beta$ = number of losses +1. Uncertainty was defined as the variance of this beta distribution: $\frac{(\alpha*\beta)^2}{\alpha+\beta+1}$ . Stimulus novelty was determined by taking the variance of a different beta distribution, where $\alpha$ = the number of times participants had seen the choice option before throughout the entire task +1 and $\beta$ = 1.

To address how these choice features differentially influenced exploration across development, we computed the differences in expected value, uncertainty, and novelty between the left and right

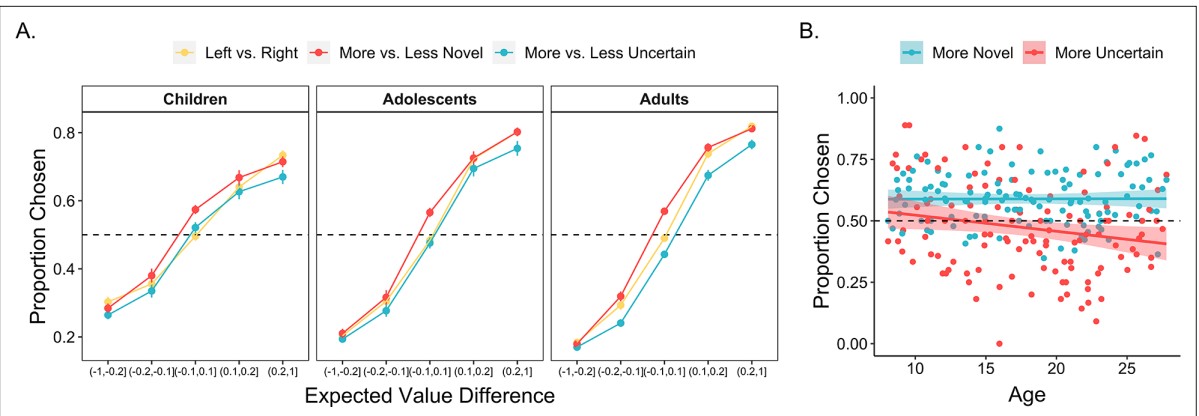

**Figure 3.** Influence of expected value, uncertainty, and novelty on choice behavior across age. (**A**) The proportion of all trials in which participants (n = 122) chose the left, more novel, and more uncertain choice option as a function of the expected value difference between the options. Participants were more likely to choose options with greater expected value, higher novelty, and lower uncertainty ($ps < 0.001$). The influence of novelty did not vary across age, whereas uncertainty was more aversive in older participants ($p < 0.001$). Points indicate age group means and error bars show standard errors. (**B**) The proportion of similar-expected-value trials (difference between the two options <0.05) in which participants chose the more novel and more uncertain option, plotted as a function of continuous age. The lines show the best-fitting linear regression lines and the shaded regions around them represent 95% confidence intervals.

choice options on every trial. We then ran a mixed-effects logistic regression examining how these differences — as well as their interactions with continuous age — related to the probability that the participant chose the left option on every trial. Participants were more likely to select the options that they had learned were more valuable, OR = 3.22 [2.82, 3.67], $\chi^2$ (1) = 155.8, $p < 0.001$. However, expected value was not the only driver of choice behavior. Participants also demonstrated a bias toward selecting more novel stimuli, OR = 1.34 [1.28, 1.41], $\chi^2$ (1) = 104.2, $p < 0.001$, and a bias *away* from choosing options with greater uncertainty, OR = 0.89 [0.83, 0.95], $\chi^2$ (1) = 11.9, $p < 0.001$. On trials in which the two choice options had similar-expected-values (<0.05 difference), participants selected the more novel option on 58.7% (SE = 0.8%) of trials and the more uncertain option on only 46.2% (SE = 0.9%) of trials.

The influence of expected value, novelty, and uncertainty on choice behavior each followed distinct developmental trajectories. Results from our regression model indicated that younger participants' choices were less value-driven relative to those of older participants, as reflected in a significant age × expected value interaction, OR = 1.22 [1.07, 1.39], $\chi^2$ (1) = 8.97, $p = 0.003$ (*Figure 3A*). These findings are consistent with a broader literature that has observed age-related improvements in the computation of expected value (*Rosenbaum and Hartley, 2019*). Importantly, however, age-related increases in these 'exploitative' choices were *not* driven by age-related differences in novelty-seeking; there was not a significant interaction between age and novelty, OR = 1.02 [0.98, 1.07], $\chi^2$ (1) = 0.96, $p = 0.327$. In contrast to the relative stability of this novelty preference across age, we observed a significant age × uncertainty interaction effect, OR = 0.89 [0.84, 0.95], $\chi^2$ (1) = 11.3, $p < 0.001$, indicating greater uncertainty aversion in older participants (*Figure 3A*). All findings held when we included block number and within-block trial number as interacting fixed effects in the model (see Appendix 1).

We further examined whether age-related increases in uncertainty aversion were due to an early *preference* to engage with uncertain options or early indifference to uncertainty. To test these possibilities, we ran an additional mixed-effects logistic regression including data only from child participants, examining how expected value, uncertainty, and novelty influenced choice. Results indicated that children's choices were significantly influenced by both expected value (OR = 2.27 [1.78, 2.91], $\chi^2$(1) = 26.8, $p < 0.001$) and novelty (OR = 1.35 [1.24, 1.48], $\chi^2$(1) = 28.4, $p < 0.001$). However, there was not an effect of uncertainty on choice, OR = 1.07 [0.945, 1.21], $\chi^2$(1) = 1.17, $p = 0.280$, indicating no significant evidence for uncertainty-seeking.

Corroborating these findings, on trials in which the two choice options had nearly identical expected values (<0.05 difference), children, adolescents, and adults, on average, selected the more novel option on 59.1% (SE = 1.7%), 59.1% (SE = 1.7%), and 58.3% (SE = 1.2%) of trials, respectively (*Figure 3B*). However, whereas adults tended to avoid the more uncertain option, selecting it on only

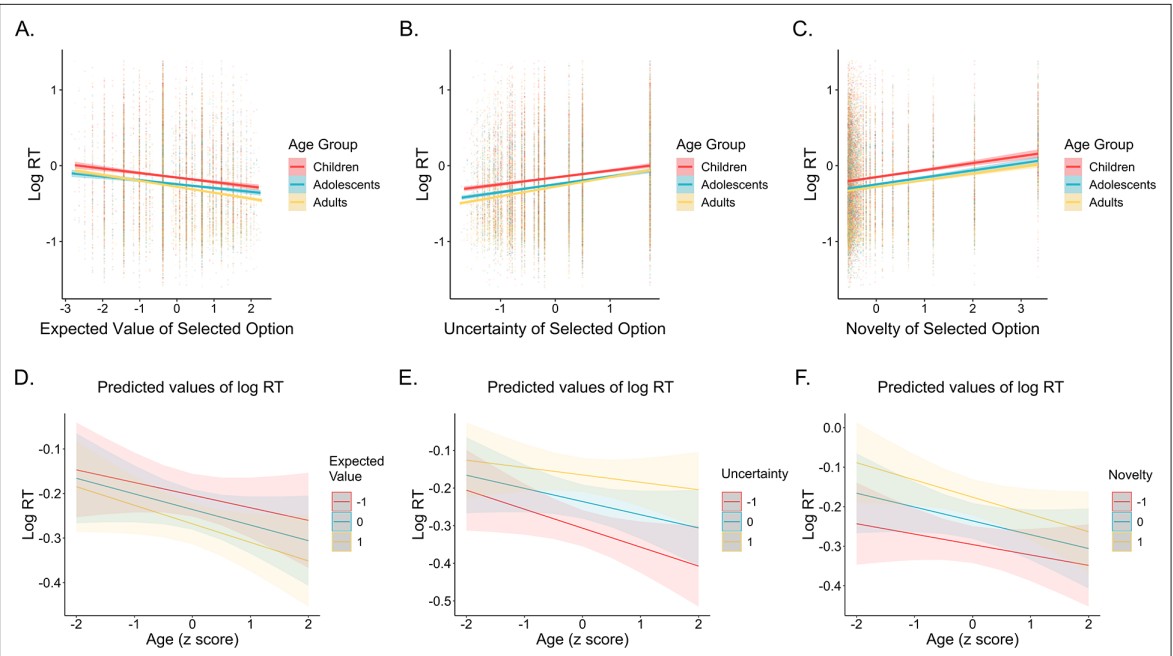

**Figure 4.** Influence of expected value, uncertainty, and novelty on choice response times across age. (**A**) Participants (n = 122) were faster to select options with higher expected values, (**B**) slower to select options with greater uncertainty, and (**C**) slower to select options with higher novelty. Individual points in panels A–C show individual log-transformed response times on each trial. The lines in panels A–C show the best-fitting linear regression lines and the shaded regions around them represent 95% confidence intervals. (**D**) The influence of expected value on response times did not vary across age, whereas younger participants demonstrated (**E**) a weaker influence of uncertainty on response times ($p = 0.005$) and (**F**) a stronger influence of novelty on response times ($p = 0.043$). The lines in panels D–E show predictions from a linear mixed-effects model and the shaded regions around them represent 95% confidence intervals.

41.9% (SE = 1.2%) of equal-expected-value trials, adolescents and children selected the more uncertain option on 48.9% (SE = 1.7%) and 52.7% (SE = 1.7%) of these trials, respectively (*Figure 3B*). Thus, taken together, these results suggest that age-related decreases in exploratory choices were driven by an increase in aversion to reward uncertainty with increasing age.

## Age-related change in sensitivity versus aversion to uncertainty

There are two potential accounts of the observed increase in uncertainty aversion with increasing age: Children may have been sensitive to reward uncertainty but not averse to it, or, they may have failed to track uncertainty at all due to the computational demands of estimating the variance of outcome distributions across the task. To disentangle these two possibilities, we examined how the uncertainty of the selected choice option influenced participant response times. If younger participants were sensitive to reward uncertainty, their response times should relate to it.

A linear mixed-effects model examining how the expected value, novelty, and uncertainty of the selected choice option — as well as their interactions with age — related to log-transformed response times revealed sensitivity to all three choice features across age (*Figure 4*). Participants responded more quickly when selecting options with higher expected values ($b = -0.03$, SE = 0.006, $F_{(1, 115.4)} = 29.7$, $p < 0.001$), and more slowly when selecting options with higher novelty ($b = 0.06$, SE = 0.004, $F_{(1, 111.3)} = 197$, $p < 0.001$) and higher uncertainty ($b = 0.07$, SE = 0.005, $F_{(1, 117.1)} = 171.7$, $p < 0.001$). We also observed significant novelty by age ($b = -0.009$, SE = 0.004, $F_{(1, 108.2)} = 4.2$, $p = 0.043$) and uncertainty by age interactions ($b = 0.015$, SE = 0.005, $F_{(1, 117.8)} = 8.2$, $p = 0.005$). Novelty had a stronger slowing influence on the response times of younger versus older participants, whereas uncertainty had a stronger slowing influence on the response times of older versus younger participants (*Figure 4*). Critically, however, we continued to observe a significant effect of uncertainty on response times when we only included children's data in the model, b = 0.04, SE = 0.01, $F_{(1, 38.2)} = 16.5$, $p < 0.001$, indicating that children's response times were sensitive to the uncertainty of the selected option (see *Appendix 1—table 5* for regression coefficients for all choice features cross age

groups). These findings held when we controlled for within-block trial number in the models ($ps <$ 0.01).

We also examined whether slower response times reflected uncertainty aversion by examining the relation between individual differences in the influence of uncertainty on choice and in the influence of uncertainty on response times. To do so, we extracted individual participants' random uncertainty slopes from a model examining how expected value, novelty, and uncertainty differences between choice options related to decisions, and individual participants' random uncertainty slopes from our model examining how the expected value, novelty, and uncertainty of the selected choice option related to response times. We then ran a linear regression examining how the effect of uncertainty on response times, age, and their interaction related to uncertainty aversion. We observed a significant negative relation between the effect of uncertainty on choice and the effect of uncertainty on response times, $\beta = -0.22$, SE = 0.09, $t = -2.5$, $p = 0.014$, indicating that participants who were more uncertainty averse were also slower to select more uncertain options. We did not observe a significant age x uncertainty interaction effect ($p = 0.81$). Thus, taken together, our findings suggest that despite not demonstrating uncertainty aversion in their decisions, children *were* sensitive to the relative reward uncertainty of different choice options throughout the task.

## Computational characterization of choice

As in *Cockburn et al., 2022*, we observed opposing effects of novelty and uncertainty on choice — though participants sought out novel options, they shied away from those with greater uncertainty. At first glance, these results are somewhat puzzling because novel options are *inherently* uncertain. Reinforcement learning models that formalize different algorithms for how the expected utilities of the choice options are computed across trials can provide greater insight into how novelty and uncertainty may *interact* to influence exploratory choice behavior.

We fit participant choice data with six different reinforcement learning models (see methods). Across models, we conceptualized the learning process as that of a 'forgetful Bayesian learner,' such that the expected value of each choice option is computed as the mean of a beta distribution with hyperparameters that reflect recency-weighted win and loss outcomes (*Cockburn et al., 2022*). We then modified this baseline model by adding either fully separable or interacting uncertainty and novelty biases. Specifically, beyond the baseline model, we fit three additional models in which uncertainty and novelty exerted separable influences on choice behavior: a model augmented with a *novelty bias* that adjusted the initial hyperparameters of each option's beta distribution, a model augmented with an *uncertainty bias* that added or subtracted each option's scaled uncertainty to its expected utility, and a model augmented with both biases. Corroborating our behavioral results, parameter estimates from the model with both a novelty and uncertainty bias revealed an age-consistent novelty preference but age-varying uncertainty aversion (see Appendix 1).

We additionally fit two models that account for interactions between novelty and uncertainty. Given the findings of *Cockburn et al., 2022*, we hypothesized that novelty may buffer the aversive influence of reward uncertainty. In other words, we expected that the extent to which the uncertainty of a given choice option would influence its utility would increase in relation to its familiarity. Thus, we fit two additional models (with and without a separate novelty bias) in which the uncertainty bias was 'gated' by stimulus familiarity (though the model with both a novelty bias and familiarity gate was not recoverable; see 'methods').

To test for age-related change in the way that novelty and uncertainty influenced exploratory choice, we compared model fits for these six models within each age group using a random-effects Bayesian model selection procedure with simultaneous hierarchical parameter estimation (*Piray et al., 2019*) and examined protected exceedance probabilities (PXPs), which reflect the probability that a given model in a comparison set is the most frequent, best-fitting model across participants, while controlling for differences in model frequencies that may arise due to chance.

In line with findings from *Cockburn et al., 2022*, we found that adult choices were best characterized by a model in which choice utilities took into account *interactions* between novelty and uncertainty. Specifically, adult choices were best captured by the familiarity-gated uncertainty model (PXP Familiarity Gate = 1), in which uncertainty aversion was greater for more familiar options. Despite showing weaker aversion to uncertainty relative to adults, adolescents were also best fit by this model

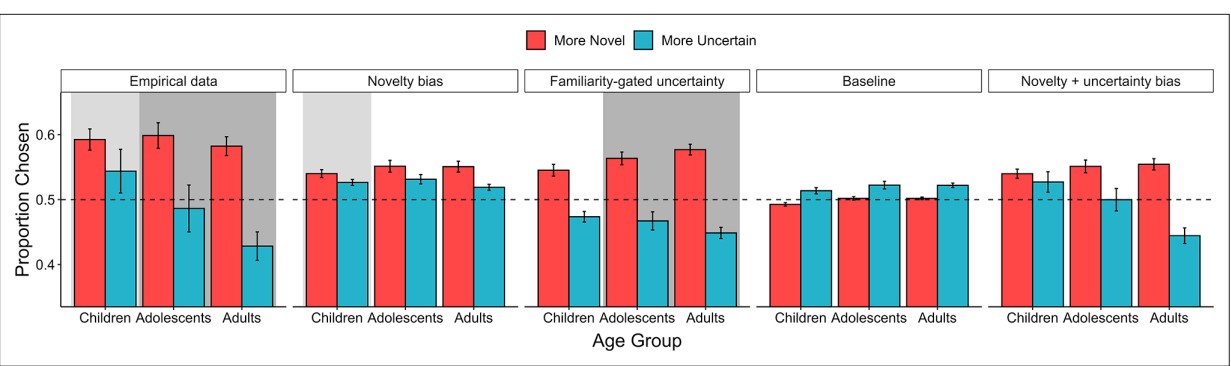

**Figure 5.** Model simulations. The average proportion of similar-expected-value trials (with expected value magnitude differences <0.05) in which both real (n = 122) and simulated (n = 122) participants chose the more novel and more uncertain option. The shaded regions show the empirical data and best-fitting model for each age group. Error bars represented the standard error across participant means. The novelty bias and familiarity-gated uncertainty model each had three free parameters, while the baseline model had two, and the novelty + uncertainty bias model had four.

(PXP Familiarity Gate = 1). Children's choices, however, were best captured by a model with a simple novelty bias (PXP Novelty Bias = 0.62; PXP Familiarity Gate = 0.38).

Parameter estimates from the winning models reflected participants' bias toward novel stimuli and away from those with high reward uncertainty. Children's average 'novelty bias' (from the group-level novelty bias model fits) was 1.49, indicating that they optimistically initiated the value of novel options. A one-sample hierarchical Bayesian inference (HBI) $t$-test examining the group-level posterior distribution of the novelty bias parameter (implemented via the cbm model-fitting package *Piray et al., 2019*), revealed that children's novelty bias was significantly different from 0, $t(16.3) = 5.96$, $p < 0.001$. The average value of the 'uncertainty bias' (from the group-level familiarity-gated uncertainty model fits) was −0.15 for both adolescents and adults. HBI $t$-tests revealed that uncertainty bias parameter estimates were significantly different from 0 in both age groups (Adolescents: $t(26.5) = −7.8$, $p < 0.001$; Adults: $t(11.3) = −4.16$, $p = 0.001$).

Model simulations revealed that the winning models well-captured qualitative features of behavioral choice data for each age group. For each model, we generated 50 simulated datasets using each of the 122 participants' trial sequence and parameter estimates (for a total of 6100 simulated agents per model). For each model, we then computed the performance of 122 simulated participants by averaging the performance of the 50 agents who shared the same trial sequence. Data from these simulations demonstrated that the familiarity-gated uncertainty model generated the most strongly diverging effects of novelty and uncertainty on choice, in line with the adult and adolescent data (*Figure 5*; also see *Appendix 1—figure 2*). The simpler novelty bias model instantiated a bias toward *both* novel and uncertain choices. Thus, these modeling results suggest that whereas adults and adolescents were more strongly deterred by the uncertainty of familiar options versus novel ones, children employed a simpler learning algorithm in which they optimistically initialized the value of novel choice options.

## Discussion

In this study, we investigated how novelty and uncertainty influence exploration across development. Though new choice options tend to have *both* high novelty and high reward uncertainty, we found that the influence of these features on decision making follow distinct developmental trajectories. While participants across age demonstrated a similar bias toward selecting more novel choice options, only older participants showed aversion to selecting those with greater uncertainty. These findings suggest that children's bias toward exploration over exploitation may arise from attenuated aversion to selecting more uncertain options rather than heightened sensitivity to novelty.

Prior studies have found that novelty may be intrinsically rewarding (*Wittmann et al., 2008*), motivating individuals to approach, learn about, and remember the new stimuli they encounter (*Houillon et al., 2013*; *Krebs et al., 2009*). Children (*Henderson and Moore, 1980*; *Mendel, 1965*; *Valenti, 1985*), adolescents (*Spear, 2000*), and adults (*Cockburn et al., 2022*; *Daffner et al., 1998*) all demonstrate novelty-seeking behavior. However, though many studies have shown novelty preferences at

different developmental stages, little work has compared novelty preferences across age. Research in rodents has suggested that adolescents may demonstrate heightened sensitivity to novelty (*Philpot and Wecker, 2008*; *Spear, 2000*; *Stansfield and Kirstein, 2006*), but to the best of our knowledge, there is no human evidence for an adolescent peak in novelty preferences. Indeed, our findings suggest that the drive to engage with novel stimuli promotes exploratory choice in a consistent manner across our age range. Moreover, the consistency of novelty-seeking across development indicates that differences in the reward value assigned to novel options cannot fully account for developmental differences in exploration.

Whereas novelty-seeking did not exhibit age-related change, uncertainty aversion increased from childhood to early adulthood, potentially reflecting developmental improvement in the strategic modulation of information-seeking. Prior studies of decision making have found that individuals across age demonstrate uncertainty aversion in some environments (*Camerer and Weber, 1992*; *Payzan-LeNestour et al., 2013*; *Rosenbaum and Hartley, 2019*) and uncertainty-seeking in others (*Blanchard and Gershman, 2018*; *Giron et al., 2022*; *Schulz et al., 2019*). These seemingly discrepant patterns of behavior may be explained by differences in the utility of resolving uncertainty across contexts. In environments where learned information can be exploited to improve subsequent choices, resolving uncertainty has high utility (*Rich and Gureckis, 2018*; *Wilson et al., 2014*), whereas in choice contexts with short temporal horizons, there is little opportunity to use learned reward information to improve future decisions (*Camerer and Weber, 1992*; *Levy et al., 2010*). In our task, individuals had a relatively short horizon over which to exploit reward probabilities that themselves required multiple trials to learn — children's reduced uncertainty aversion may have emerged from insensitivity to the limited utility of gaining additional information about the most uncertain choice options (*Somerville et al., 2017*).

Importantly, even though children's choices were not uncertainty averse, children's slower response times when engaging with more uncertain options suggests that they were able to track uncertainty. Indeed, the developmental decrease in uncertainty-seeking behavior that we observed here is in line with what has been observed in prior studies in which uncertainty was easier to discern. For example, in one study (*Blanco and Sloutsky, 2021*), children tended to select choice options with hidden reward amounts over those with visible reward amounts. In other studies with spatially correlated rewards, children could use the layout of revealed outcomes to direct their sampling toward unexplored regions (*Giron et al., 2022*; *Meder et al., 2021*; *Schulz et al., 2019*). In studies of causal learning and exploratory play, young children often use experiences of surprise or the coherence of their own beliefs about the world to direct their exploration toward uncertain parts of the environment (*Bonawitz et al., 2012*; *Schulz and Bonawitz, 2007*; *Wang et al., 2021*). Here, we extend these past findings to demonstrate that children are sensitive to uncertainty even when it depends on distributions of binary outcomes.

Our observation of an influence of uncertainty on children's reaction times suggests that uncertainty did affect how children made value-based decisions. Future work could fit cognitive models to both participants' choices and response times to investigate how, across age, uncertainty influences component cognitive processes involved in value-based decision making. For example, researchers could use sequential sampling models to test different hypotheses about how value uncertainty — and its interactions with both expected value and novelty — influences both the rate at which participants accumulate evidence for a given option as well as the evidence threshold that must be reached for a response to be made (*Lee and Usher, 2023*; *Wu et al., 2022*). In addition, these approaches could be integrated with reinforcement learning models (*Fontanesi et al., 2019*) to gain further resolution into how the learned features of different options influence the choices participants make and the speed with which they make them.

Our computational modeling findings largely replicated previous work suggesting that by adulthood, novelty and uncertainty interact competitively (*Cockburn et al., 2022*), exerting opposing motivational influences on decision-making. Using functional magnetic resonance imaging (fMRI), this prior adult study (*Cockburn et al., 2022*) revealed that activation in the ventral striatum reflects a biased reward prediction error consistent with optimistic value initialization for novel stimuli, whereas activation in the medial prefrontal cortex (mPFC) reflects the subjective utility of uncertainty reduction. In line with choices being best characterized by a model in which the aversive influence of uncertainty was dampened by novelty, the integration of uncertainty into these mPFC value representations

was similarly diminished for novel stimuli (*Cockburn et al., 2022*). *Cockburn et al., 2022* posited that attenuated uncertainty aversion for novel stimuli may promote exploration even in environments where deriving the prospective utility of uncertainty reduction is difficult. Our findings further suggest that these competitive interactions between uncertainty and novelty may emerge in adolescence, as connectivity between cortical and subcortical circuitry matures (*Casey et al., 2019*; *Parr et al., 2021*). However, more neuroimaging work is needed to unveil how changes in neural circuitry support the use of these choice features during decision making across development.

Recent theories have proposed that heightened exploration is an adaptive quality of childhood (*Giron et al., 2022*; *Gopnik, 2020*), but these theoretical accounts — and the empirical work they have inspired — have left open the question of how different features of the environment elicit this strong, exploratory drive early in life. Here, we demonstrated that the developmental shift from more exploratory to more exploitative behavior may arise from strengthening aversion to selecting options with higher reward uncertainty with increasing age, rather than from changes in novelty-seeking. Importantly, in the real world, exploration manifests through interaction with dynamic ecological contexts in which novelty and uncertainty may be highly idiosyncratic. Compared to adolescents and adults, children may encounter a greater number of novel options due to their relative lack of life experience. As they gain more autonomy, adolescents may find themselves facing more decisions (particularly in the social domain) with unknown reward outcomes. The current findings help build toward a comprehensive understanding of developmental change in exploration by disentangling the cognitive processes that govern how individuals interact with these core features of the choice options present in natural environments.

## Methods
### Participants
One hundred and twenty-two participants between the ages of 8 and 27 years old (mean age = 17.9 years, SD age = 5.6 years, 62 females, 59 males, 1 non-binary) completed the experiment. Based on prior, similar studies of value-guided decision making from childhood to adulthood (*Habicht et al., 2021*; *Somerville et al., 2017*), we determined a target sample size of $N$ = 120, evenly distributed across our age range, prior to data collection. The final analyzed sample of 122 participants comprised $n$ = 30 children (mean age = 10.5 years; SD age = 1.4 years; range = 8.1–12.7 years; 14 females), $n$ = 30 adolescents (mean age = 15.5 years; SD age = 1.3 years; range = 13.6–17.8 years; 16 females), and $n$ = 62 adults (mean age = 22.6 years; SD age = 2.8 years; range = 18.1–27.8 years; 32 females). Data from one additional participant were not analyzed because the participant chose to stop the experiment prior to completing the entire exploration task. Two participants included in the final analyzed sample were excluded from memory test analyses due to technical errors during data acquisition.

Participants were recruited from the local New York City community. Participants reported normal or corrected-to-normal vision and no history of diagnosed psychiatric or learning disorders. Based on self- or parental report, 33.6% of participants were Asian, 33.6% were White, 18.0% were two or more races, 13.1% were Black, and 1.6% were Pacific Islander/Native Hawaiian. In addition, 15.6% of participants were Hispanic.

Research procedures were approved by New York University's Institutional Review Board (IRB-2016-1194 and IRB-FY2021-5356). Adult participants provided written consent prior to participating in the study. Children and adolescents provided written assent, and their parents or guardians provided written consent on their behalf, prior to their participation. All participants were compensated $15/hr for the experimental session. Participants were told that they would receive an additional bonus payment based on their performance in the experiment; in reality, all participants received an additional $5 bonus payment.

### Task
#### Exploration task
Participants completed a child-friendly decision-making task adapted from one used in a prior adult study (*Cockburn et al., 2022*). The child-friendly version of the task was framed within an 'Enchanted Kingdom' narrative and included fewer stimuli and trials per block than the version used in prior work (*Cockburn et al., 2022*) with adults. Within this narrative framework, participants were tasked with

finding gold coins to raise money to build a bridge to unite two sides of an enchanted kingdom. Various creatures had hidden the gold coins in different territories around the kingdom. On every trial, participants had to choose between two hiding spots to search for a coin.

The task was divided into 10 blocks of 15 trials. Each block took place within a different territory in which coins were hidden by a distinct creature. Each creature hid their coins among three possible locations. Half of the blocks were 'easy' — in easy blocks, the creature's favorite hiding spot held a coin on 80% of trials, their second favorite held a coin on 50% of trials, and their least favorite held a coin on 20% of trials. The other half of blocks were 'hard' — in hard blocks, the creature's favorite hiding spot held a coin on 70% of trials, their second favorite held a coin on 50% of trials, and their least favorite held a coin on 30% of trials. Participants were not explicitly informed of these probabilities, and had to learn, through trial and error, where each creature was most likely to hide a coin.

On every trial, participants viewed two hiding spots and had 4 s to select one in which to search for a coin by pressing one of two keys on a standard keyboard (*Figure 1*). After a brief delay in which the option they selected was outlined (500 ms), participants saw the outcome of their choice — either a coin or an X indicating that they had not found a coin (1.5 s). Throughout each block, the background of the screen indicated the territory and a picture in the lower left corner indicated the creature that had hidden coins there.

Importantly, after the first block, each subsequent block contained two hiding spots that participants had encountered in previous blocks and one novel hiding spot they had not seen before. Though participants had already encountered two of the hiding spots within each block, the reward probabilities were re-randomized for every creature. In this way, the task dissociated sensory *novelty* and reward *uncertainty*. At the beginning of every block, the novelty of each hiding spot varied — at least one hiding spot was completely novel, whereas from the second block on, the other two had been encountered anywhere from 4 to 82 times (mean = 22.4 encounters, SD = 13.8 encounters) — but *all* hiding spots had high reward uncertainty. Participants were explicitly told that the reward probabilities were reset in every block; within the task narrative, this was framed as each creature having different favorite hiding spots in their respective territory (see Appendix 1 for analyses demonstrating that participants of all ages indeed comprehended these instructions and 'reset' the reward probabilities at the beginning of each block).

The order of the creatures and the hiding spots assigned to each creature were randomized for each participant. Within each block, the reward probabilities assigned to the two 'old' hiding spots and the novel hiding spot were randomized. On each trial, the two hiding spots that appeared as choice options and their positions on the screen (left or right) were also randomized.

### Memory test

Immediately after the exploration task, participants completed a surprise memory test. They were shown each of the ten creatures, one at a time, and asked to select its favorite hiding spot from an array of five options, using numbers on the keyboard. The array of five options always included the hiding spot in which the creature was most likely to hide the coin (the correct answer), the two other hiding spots where that creature hid coins, a previously encountered hiding spot from a different block of the task, and a new hiding spot that was not presented in the exploration task.

### Instructions and practice

Prior to completing both the exploration and the memory task, participants went through extensive, interactive instructions with an experimenter. The instructions were written and illustrated on the computer screen, and an experimenter read them aloud. During the instructions, participants were informed that (1) hiding spots may repeat throughout the task but each creature had different favorite hiding spots, (2) a creature would not *always* hide its coins in its favorite spot, and (3) each creature's hiding spot preferences remained stable throughout the entire block. They also discussed three comprehension questions with an experimenter to ensure their full understanding of the task; comprehension questions were not scored, but instead remained on the screen until participants selected the correct answer (with experimenter guidance, if needed). Finally, participants completed two full practice blocks with stimuli that were not used in the main task.

### WASI

After the exploration task and memory test, participants were administered the vocabulary and matrix reasoning subtests of the Wechsler Abbreviated Scale of Intelligence (*Wechsler, 2011*). Because our primary aim was to address the relation between age and exploratory behavior, we report results from the models *without* WASI scores in the main text of the manuscript and include results from the models with WASI scores in Appendix 1.

## Analysis approach

### Treatment of age

We treated age as a continuous variable in all regression analyses. We binned participants into three age groups (children aged 8–12 years, adolescents aged 13–17 years, and adults aged 18–27 years) for data visualization and model comparison purposes. Our adult age group spanned a greater age range and included double the number of participants as our child and adolescent age groups; as such, we include additional visualizations and model comparison results that subdivide this group into college-aged adults (18–22 years) and post-college-aged adults (23–27 years) in Appendix 1.

Because we originally hypothesized that the influence of expected value, uncertainty, and novelty would change monotonically with age, we included linear age in all regression analyses. To account for potential non-linearities in our age effects, we report results from analyses including quadratic age in Appendix 1, though we note that the quadratic age effects we observed do not hold when we control for potential cohort-level IQ differences across our sample (see Appendix 1).

### Mixed-effects modeling methods

Data processing was conducted in both Matlab 2020b (*The MathWorks Inc, 2020*) and R version 4.1.1 (*R Development Core Team, 2018*). Mixed-effects models were run using the 'afex' package (*Singmann et al., 2020*). Continuous variables were *z*-scored across the entire dataset prior to their inclusion in the models. Choice trials in which participants failed to make a response, or responded in faster than 200ms were excluded from all analyses ($n$ = 154 out of 18,300 total trials; max 13 out of 150 trials excluded per participant). Models included random participant intercepts and slopes across all fixed effects, except where noted due to convergence failures. For logistic mixed-effects models, we assessed the significance of fixed effects via likelihood ratio tests. For linear mixed-effects models, we assessed the significance of fixed effects via F tests using the Satterthwaite approximation to determine the degrees of freedom.

## Computational modeling

We fit participant data with six variants of a reinforcement learning model that reflected different hypotheses about how expected value, novelty, and uncertainty may influence the estimated utility of the choice options presented on each trial (*Cockburn et al., 2022*). Across all models, these estimated utilities ($V$) were transformed into choice probabilities via a softmax function, with an inverse temperature parameter ($\beta_{SM}$) that captured the extent to which estimated utilities drove participant choices, with higher estimates reflecting more value-driven choices, and lower estimates reflecting greater choice stochasticity such that:

$$p\left(c_t = 1\right) = \frac{1}{1 + e^{\beta_{SM} * (V_2 - V_1)}}$$

where $c_t$ is the choice they made on trial $t$.

Following the approach of *Cockburn et al., 2022*, we modeled the task from the perspective of a forgetful Bayesian learner. From this perspective, each option is represented as a Beta distribution, where the two parameters that define the distribution ($\alpha$ and $\beta$) reflect recency-weighted win and loss histories. The recency weights are governed by a learning rate-free parameter ($\eta$), with higher values reflecting greater weight placed on more recent outcomes such that:

$$\alpha_i = 1 + \sum_{t=0}^{T-1} \eta^{T-t} * O_t^w$$

and

$$\beta_i \;=\; 1 \;+\; \sum_{t=0}^{T-1} \eta^{T-t} \;*\; O_t^L$$

where $T$ represents the current trial within the block, and $O_t^W$ and $O_t^L$ represent win and loss outcomes on each trial (such that they are set to 1 for win and loss outcomes, and 0 otherwise, respectively). In line with our model-free analyses, we conceptualize the mean of the beta distribution defined by these two parameters as the choice option's expected value ($Q$) and its variance as the option's uncertainty ($U$).

*Baseline model (two parameters).* The baseline model captured the hypothesis that choices were driven by each option's expected value. As such, in the two-parameter baseline model, the means of the beta distributions determined the utility of each choice option.

*Novelty bias model (three parameters).* The novelty bias model was identical to the baseline model except that a novelty bias was incorporated by inflating the initial values of the hyperparameters of the beta distribution of each choice option on its first appearance in the task. This bias, implemented as free parameter, $N$, inflated either $\alpha$ (reflecting optimistic initialization or novelty-seeking behavior) or $\beta$ (reflecting pessimistic initialization, or novelty aversion).

*Uncertainty bias model (three parameters).* The uncertainty bias model was identical to the baseline model except that each option's utility was defined as the *sum* of its expected value and weighted uncertainty. The weight placed on the uncertainty of each option ($U$), implemented as free parameter, $w_U$, could be negative (reflecting uncertainty aversion) or positive (reflecting uncertainty-seeking).

*Novelty and uncertainty bias model (four parameters).* The novelty and uncertainty bias model incorporated both the novelty bias and uncertainty bias parameters described above.

*Familiarity-gated uncertainty model (three parameters).* This model captured the hypothesis that novelty and uncertainty *interacted* to influence the subjective utility of the choice options. The model implemented a 'familiarity-gating' mechanism, such that the uncertainty of *less novel* options influenced their subjective utility to a greater extent (***Cockburn et al., 2022***). As in our model-free analyses, we mathematically defined each option's novelty as the variance of a beta distribution with hyperparameters $\alpha$ = 1 + number of times each option has been presented and $\beta$ = 1. We then defined option familiarity, $F$, as 1 − option novelty. Finally, we then further multiplied each option's weighted uncertainty value by option familiarity such that:

$$V_i \;=\; Q_i \;+\; F_i \;*\; U_i \;*\; w_u$$

*Novelty-biased familiarity-gated uncertainty model (four parameters).* Finally, this model was identical to the familiarity-gated uncertainty model, but it also implemented optimistic or pessimistic initialization of novel choice options based on each participant's novelty bias, $N$.

## Model-fitting methods

Computational modeling was conducted using the computational and behavioral modeling (cbm) package (***Piray et al., 2019***) within Matlab 2020b (***The MathWorks Inc, 2020***). We first fit each of the six model's to each participant's choice data, using common priors $(N(\mu = 0, \sigma^2 = 6.25))$ (***Cockburn et al., 2022***). Because the package relies on normally distributed parameters, following the approach of ***Cockburn et al., 2022***, across all models, we first transformed the inverse temperature parameter, which was constrained to be between 0 and 20, and the learning rate, which was constrained to be between 0 and 1, using sigmoidal functions.

These first-level fits were then fed into a second-level fitting and model comparison algorithm. Here, we performed fitting and model comparison separately for each age group to determine the most frequent best-fitting model for children, adolescents, and adults. The second-level fitting procedure performs simultaneous hierarchical parameter estimation and Bayesian model comparison, in which each participant is treated as a random effect (i.e., different participants may be best fit by different models). Through this approach, the extent to which the parameter estimates for a particular participant influences the group-level empirical prior over each parameter is determined by the extent to which their choice data are best captured by a particular model (***Piray et al., 2019***).

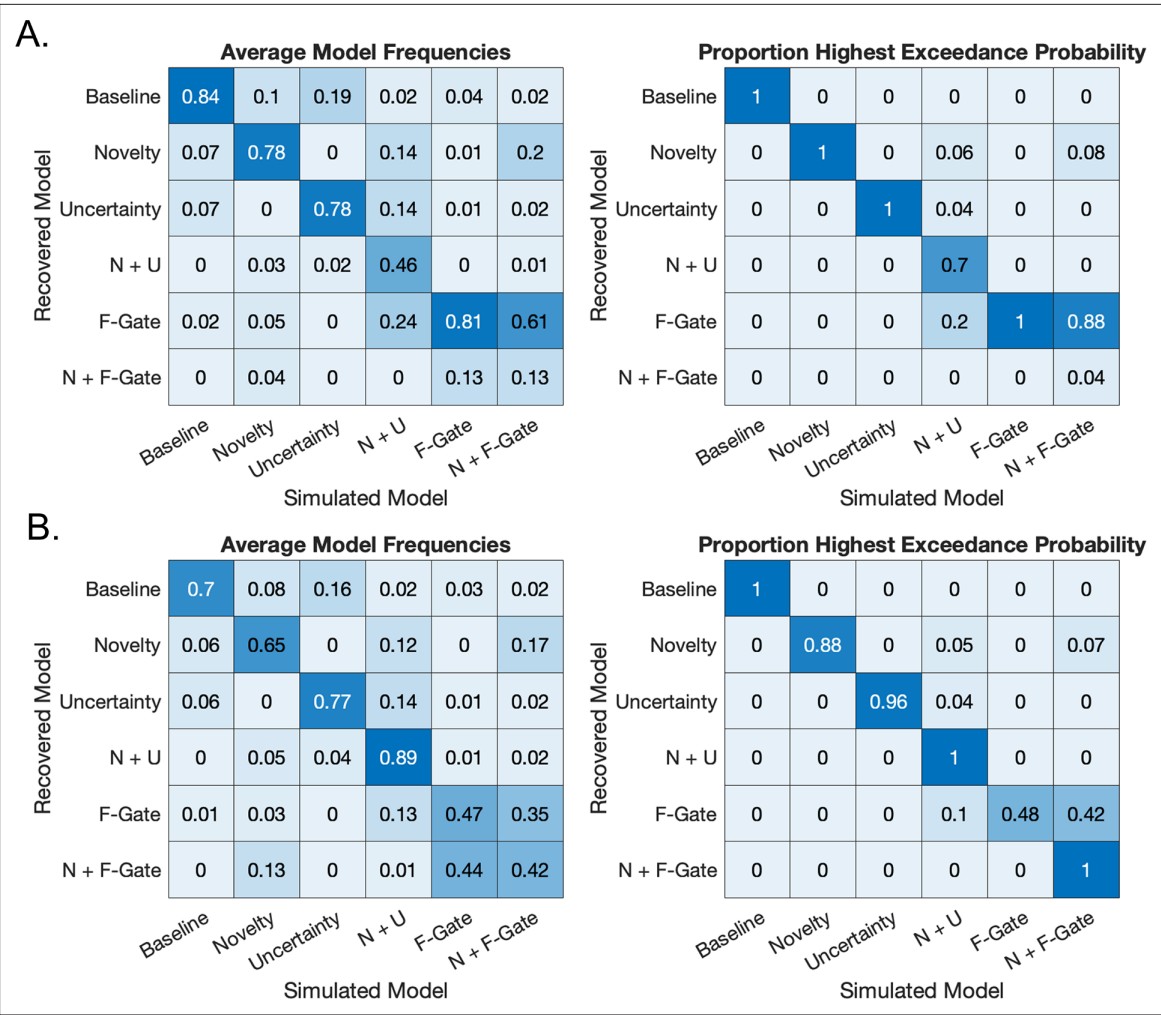

**Figure 6.** Model recovery results. (**A**) Confusion matrices showing the probability of each fitted model given a simulated model. Across simulated datasets, the most frequent, best-fitting model usually matched the model that was used to generate the data. (**B**) Inversion matrices showing the probability of each simulated model given a fitted model. Together, these results indicate that the familiarity-gated uncertainty model could not be distinguished from the familiarity-gated uncertainty model with an additional novelty parameter.

## Model recovery

To determine the extent to which the six models in our comparison set were identifiable, we generated 50 simulated datasets using each participants' trial sequence and parameter estimates for each model. We then fit each of these datasets with all six models, using the same two-stage fitting process that we used for our empirical data: For each of the 50 simulated datasets, we first fit each of the six models to each simulated participant's choice data, using common priors $(N(\mu = 0, \ \sigma^2 = 6.25))$. We then fed these fits into a second-level fitting and model comparison algorithm. For each simulated model, we examined the average proportion of participants best fit by each of the six models (*Figure 6*). We also examined the proportion of the 50 datasets simulated with each model for which each fitted model was the most frequent best-fitting model, according to exceedance probabilities (*Figure 6*).

These results revealed that in general, model identifiability was strong, particularly for the baseline, novelty bias, uncertainty bias, and familiarity-gated uncertainty model. Model identifiability for the novelty and uncertainty bias model was weaker, likely because participants may have had novelty- or uncertainty-bias parameter values that were close to 0, leading their data to be better fit by a model that implemented only one of the two biases. Finally, the familiarity-gated uncertainty model with an additional novelty bias (implemented via asymmetric value initialization) was not recoverable at all and was almost always confused for the familiarity-gated uncertainty model. By attenuating the (generally

aversive) influence of uncertainty on the utility of more novel stimuli, the familiarity-gated uncertainty model inherently implements a bias toward more novel options; thus, our task does not have the resolution to effectively determine whether there may also be an *additional* bias toward novelty instantiated via optimistic value initialization. However, the model recoverability results demonstrate that models that implement an interactive effect of novelty and uncertainty can be clearly distinguished from those that do not. Moreover, these model recoverability results indicate that the novelty bias model and familiarity-gated uncertainty model were highly distinguishable from one another, supporting our claim that children employed a value-updating mechanism that was fundamentally distinct from the one used by adolescents and adults in our task.

## Acknowledgements

We thank May Levin for help with task design. This work was supported by a National Science Foundation CAREER Award (Grant No. 1654393 to CAH), the NYU Vulnerable Brain Project (grant to CAH), the Department of Defense (NDSEG fellowship to KN), the National Institute of Mental Health (F31 MH129105 to KN; T32 MH019524 to REM), the National Institute of Child Health and Human Development (F31 HD097873 to REM), the Leon Levy Fellowship in Neuroscience (to REM), and the National Institute on Drug Abuse (F32 DA047047 to GMR). This work was supported in part through the NYU IT High Performance Computing resources, services, and staff expertise.

## Additional information

### Competing interests

Catherine A Hartley: Reviewing editor, eLife. The other authors declare that no competing interests exist.

### Funding

| Funder | Grant reference number | Author |
|---|---|---|
| National Science Foundation | 1654393 | Catherine A Hartley |
| NYU Vulnerable Brain Project | | Catherine A Hartley |
| U.S. Department of Defense | NDSEG Fellowship | Kate Nussenbaum |
| National Institute of Mental Health | F31 MH129105 | Kate Nussenbaum |
| National Institute of Mental Health | T32 MH019524 | Rebecca E Martin |
| Eunice Kennedy Shriver National Institute of Child Health and Human Development | F32 HD097873 | Rebecca E Martin |
| Leon Levy Foundation | Fellowship in Neuroscience | Rebecca E Martin |
| National Institute on Drug Abuse | F32 DA047047 | Gail M Rosenbaum |
| National Institute of Mental Health | R01MH126183 | Catherine A Hartley |

The funders had no role in study design, data collection, and interpretation, or the decision to submit the work for publication.

### Author contributions

Kate Nussenbaum, Data curation, Software, Formal analysis, Validation, Investigation, Visualization, Methodology, Writing – original draft, Project administration; Rebecca E Martin, Conceptualization,

Software, Formal analysis, Investigation, Methodology, Project administration, Writing – review and editing; Sean Maulhardt, Yi (Jen) Yang, Greer Bizzell-Hatcher, Naiti S Bhatt, Maximilian Koenig, Investigation, Writing – review and editing; Gail M Rosenbaum, Software, Investigation, Writing – review and editing; John P O'Doherty, Conceptualization, Supervision, Methodology, Writing – review and editing; Jeffrey Cockburn, Conceptualization, Software, Methodology, Writing – review and editing; Catherine A Hartley, Conceptualization, Supervision, Funding acquisition, Project administration, Writing – review and editing

**Author ORCIDs**
Kate Nussenbaum (ID) https://orcid.org/0000-0002-7185-6880
Catherine A Hartley (ID) https://orcid.org/0000-0003-0177-7295

**Ethics**
Research procedures were approved by New York University's Institutional Review Board (IRB-2016-1194 and IRB-FY2021-5356). Adult participants provided written consent prior to participating in the study. Children and adolescents provided written assent, and their parents or guardians provided written consent on their behalf, prior to their participation. All participants were compensated $15/hr for the experimental session.

**Decision letter and Author response**
Decision letter https://doi.org/10.7554/eLife.84260.sa1
Author response https://doi.org/10.7554/eLife.84260.sa2

## Additional files

**Supplementary files**
• MDAR checklist

**Data availability**
The study task code, data, and analysis code are publicly accessible on the Open Science Framework: https://osf.io/cwf2k/.

The following dataset was generated:

| Author(s) | Year | Dataset title | Dataset URL | Database and Identifier |
|---|---|---|---|---|
| Nussenbaum K, Martin R, Maulhardt S, Yang Y, Bizzell-Hatcher G, Bhatt N, Scheuplein M, Rosenbaum G, O'Doherty J, Cockburn J, Hartley C | 2022 | Data from: Novelty and uncertainty differentially drive exploration across development | https://doi.org/10.17605/OSF.IO/CWF2K | Open Science Framework, 10.17605/OSF.IO/CWF2K |

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

# Appendix 1

## Supplementary information

### Choice feature regression with additional covariates

In the main text of the manuscript, we examined the influence of three choice features — expected value, novelty, and uncertainty — as well as their interactions with age on choice behavior. Specifically, we computed the difference in feature values between the left and right choice option on each trial and ran a mixed-effects logistic regression with 'left choice' (coded as 0 or 1) as the dependent variable. Below, we describe results of augmented versions of this model that explore the influence of additional covariates on choice behavior.

### Effects of within-block trial on choice behavior

We examined how the influence of each choice feature changed over the course of learning within each block by including within-block trial number as an interacting fixed effect in our mixed-effects logistic regression (*Appendix 1 — table 1*). We continued to observe robust effects of expected value, novelty, and uncertainty (*p*s < 0.001), as well as age x expected value and age x uncertainty interaction effects (*p*s < .003). Here, we also observed an interaction between expected value and trial number, such that participant choices became less value-driven throughout the task block (*p* < .001). Participant choices also became more novelty-seeking, as indicated by a novelty x trial number interaction effect (*p* = .041). We did not observe a significant uncertainty x trial number interaction effect (*p* = .858). In addition, we did not observe any significant three-way interactions between trial number, age, and choice features (*p*s > .14).

**Appendix 1—table 1.** Influences on exploratory choice including within-block trial number.

|  | Odds ratio | 95% confidence interval | $X^2$ | *p* |
|---|---|---|---|---|
| Intercept | 1.00 | [.95, 1.05] |  |  |
| Expected Value | 3.39 | [2.98, 3.86] | 164.5 | <0.001 |
| Uncertainty | 0.87 | [0.81, 0.93] | 15.6 | <0.001 |
| Novelty | 1.38 | [1.31, 1.46] | 107.1 | <0.001 |
| Expected Value × Trial | 0.82 | [0.78, 0.86] | 72.5 | <0.001 |
| Uncertainty × Trial | 1.00 | [0.96, 1.05] | 0.03 | 0.858 |
| Novelty × Trial | 1.03 | [1.00, 1.07] | 4.2 | 0.041 |
| Expected Value × Age | 1.23 | [1.08, 1.40] | 9.8 | 0.002 |
| Uncertainty × Age | 0.88 | [0.83, 0.94] | 14.1 | <0.001 |
| Novelty × Age | 1.04 | [0.98, 1.09] | 1.8 | 0.185 |
| EV × Trial × Age | 1.00 | [0.96, 1.05] | 0.0 | 0.906 |
| Uncertainty × Trial × Age | 1.02 | [0.98, 1.06] | 0.7 | 0.409 |
| Novelty × Trial × Age | 1.02 | [0.99, 1.06] | 2.1 | 0.145 |

### Effects of expected value, novelty, and uncertainty on choice behavior across age, controlling for age-normed WASI scores

As noted in the main text of the manuscript, participants were administered the vocabulary and matrix reasoning subtests of the Wechsler Abbreviated Scale of Intelligence (*Wechsler, 2011*). We followed the standard procedure to compute age-normed IQ scores for each participant based on their performance on these two subtests. Two participants (one child and one adolescent) did not complete the WASI and therefore were excluded from all analyses involving WASI scores. We observed a significant relation between participant age and age-normed WASI scores, $\beta = -0.18$, SE = 0.09, *t* = −1.98, p = 0.05, indicating that relative to other individuals their age, younger participants in our sample had stronger fluid reasoning abilities than older participants in our sample.

To account for potential effects of reasoning differences on exploratory choice behavior, we re-ran our choice feature model but additionally included WASI scores — as well as all two-way interactions between WASI scores and choice features, and all three-way interactions between WASI scores, age,

and choice features — as fixed effects (*Appendix 1—table 2*). Here, we continued to observe robust value-seeking, novelty-seeking, and uncertainty aversion ($ps < 0.001$). We also continued to observe age × expected value and age × uncertainty interaction effects ($ps < 0.001$). We further observed an expected value × WASI score interaction effect ($p < 0.001$), indicating that individuals with stronger fluid reasoning abilities also demonstrated more value-seeking choice behavior. Finally, we observed a novelty × WASI score interaction effect ($p = 0.025$), indicating that individuals with higher WASI scores also demonstrated stronger novelty-seeking. No other effects or interactions were significant ($ps > 0.14$).

**Appendix 1—table 2.** Influences on exploratory choice including WASI scores.

|  | Odds ratio | 95% confidence interval | $X^2$ | $p$ |
|---|---|---|---|---|
| Intercept | 1.00 | [0.95, 1.05] |  |  |
| Expected Value | 3.31 | [2.92, 3.75] | 167.4 | <0.001 |
| Uncertainty | 0.87 | [0.81, 0.93] | 16.0 | <0.001 |
| Novelty | 1.35 | [1.29, 1.42] | 104.0 | <0.001 |
| Expected Value × Age | 1.29 | [1.14, 1.46] | 15.8 | <0.001 |
| Uncertainty × Age | 0.89 | [0.83, 0.95] | 12.6 | <0.001 |
| Novelty × Age | 1.04 | [0.99, 1.09] | 2.1 | 0.145 |
| Expected Value × WASI | 1.33 | [1.17, 1.50] | 19.1 | <0.001 |
| Uncertainty × WASI | 0.96 | [0.90, 1.02] | 1.5 | 0.217 |
| Novelty × WASI | 1.06 | [1.01, 1.11] | 5.0 | 0.025 |
| EV × Age × WASI | 1.06 | [0.94, 1.19] | 1.1 | 0.288 |
| Uncertainty × Age × WASI | 0.96 | [0.90, 1.02] | 2.0 | 0.158 |
| Novelty × Age × WASI | 1.01 | [0.97, 1.06] | 0.4 | 0.532 |

## Effects of expected value, novelty, uncertainty, and block number on choice behavior across age

We additionally examined how the influence of each choice feature changed over the course of the experiment by including interactions between choice features and block number in our mixed-effects logistic regression (*Appendix 1—table 3*). We did not observe a significant effect of block number, nor did the influence of any choice feature vary across blocks.

**Appendix 1—table 3.** Influences on exploratory choice including block number.

|  | Odds ratio | 95% confidence interval | $X^2$ | $p$ |
|---|---|---|---|---|
| Intercept | 1.00 | [0.96, 1.05] |  |  |
| Expected Value | 3.22 | [2.82, 3.67] | 155.6 | <0.001 |
| Uncertainty | 0.88 | [0.83, 0.94] | 12.3 | <0.001 |
| Novelty | 1.35 | [1.29, 1.41] | 105.1 | <0.001 |
| Expected Value × Age | 1.22 | [1.07, 1.39] | 9.0 | 0.003 |
| Uncertainty × Age | 0.89 | [0.83, 0.95] | 11.4 | <0.001 |
| Novelty × Age | 1.02 | [0.98, 1.07] | 1.0 | 0.330 |
| Expected Value × Block | 0.99 | [0.95, 1.03] | 0.2 | 0.666 |
| Uncertainty × Block | 1.02 | [0.98, 1.06] | 0.7 | 0.398 |
| Novelty × Block | 0.98 | [0.95, 1.02] | 1.0 | 0.310 |
| EV × Age × Block | 1.00 | [0.96, 1.04] | 0.0 | 0.913 |

*Appendix 1—table 3 Continued on next page*

*Appendix 1—table 3 Continued*

|  | Odds ratio | 95% confidence interval | $X^2$ | *p* |
|---|---|---|---|---|
| Uncertainty × Age × Block | 1.01 | [0.97, 1.05] | 0.3 | 0.580 |
| Novelty × Age × Block | 1.00 | [0.96, 1.03] | 0.0 | 0.924 |

## Ensuring reward probabilities were 'reset' in each block

Participants were instructed that the reward probabilities associated with each choice option were reset at the beginning of each block — within the narrative framework of the task, this was described as each creature having different favorite hiding spots. It is possible, however, that participants carried over learned value information from prior blocks. To test whether this was the case, we additionally computed the expected value of each choice option on each trial by taking the means of the beta distribution defined by the number of wins and losses that the participant experienced for each option *over the entire task*, rather than just within the current block. We then examined whether this expected value variable — which we refer to as the 'non-reset expected value' — better explained participant choices than the appropriately reset expected value variable.

At the group level, the model with 'non-reset expected value' provided a worse fit to the data than the original model with expected value (Akaike Information Criterion [AIC] non-reset: 20,032, AIC reset: 18,215). Thus, this suggests that participants did indeed treat the reward probabilities as being reset at the beginning of each block, rather than carrying over learned value information from previous blocks. Further, to ensure that participants across the entire age range appropriately reset the reward probabilities associated with familiar choice options, we compared models with each expected value term within each age group separately. In the models run separately for each age group, we removed age interaction effects. As at the group level, within each age group, the models with the non-reset expected value term provided a worse fit to the data (*Children*: AIC non-reset: 5170; AIC reset: 4880; *Adolescents*: AIC non-reset: 4976; AIC reset: 4516, *Adults*: AIC non-reset: 9900; AIC reset: 8829). These findings suggest that across age, participants treated familiar stimuli that appeared in each new block of the task as having new reward probabilities, in line with the task manipulation and the explicit instructions they received.

However, though these results indicate a stronger effect of reset values versus non-reset values on choice, participants may have still demonstrated a lingering influence of the reward history of familiar choice options. We further ran an additional mixed-effects logistic regression in which we included *both* value predictors, as well as novelty, uncertainty, and their interactions with age as fixed effects. Here, we did observe a significant effect of non-reset value on choice, OR = 1.09 [1.04, 1.15], $X^2(1)$ = 10.5, *p* = 0.001, indicating that participants' choices were biased by the rewards they experienced from familiar options in prior blocks. Critically, however, this effect did not interact with age (*p* = 0.490), and we continued to observe robust effects of reset value, novelty, and uncertainty on choice, as well as age × reset value and age × uncertainty interaction effects (*p*s < 0.006; *Appendix 1— table 4*). Given the strong effect of reset value on choice across age as well as the task's inclusion of extensive, child-friendly instructions, these results suggest that though participants understood the task's reward structure, they were still biased by the reward history of familiar options.

**Appendix 1—table 4.** Influences on exploratory choice including non-reset expected value.

|  | Odds ratio | 95% CI | $X^2$ | *p* |
|---|---|---|---|---|
| Intercept | 0.997 | [0.948, 1.05] |  |  |
| Expected Value | 3.04 | [2.65, 3.49] | 141.0 | <0.001 |
| Non-reset Expected Value | 1.09 | [1.04, 1.15] | 10.5 | 0.001 |
| Uncertainty | 0.902 | [0.842, 0.967] | 8.0 | 0.005 |
| Novelty | 1.37 | [1.31, 1.45] | 100.0 | <0.001 |
| Expected Value × Age | 1.22 | [1.06, 1.46] | 7.8 | 0.005 |
| Non-reset Expected Value × Age | 0.981 | [0.930, 1.04] | 0.5 | 0.490 |
| Uncertainty × Age | 0.884 | [0.825, 0.948] | 11.7 | <0.001 |

*Appendix 1—table 4 Continued on next page*

*Appendix 1—table 4 Continued*

| | Odds ratio | 95% CI | $X^2$ | *p* |
|---|---|---|---|---|
| Novelty × Age | 1.02 | [0.972, 1.08] | 0.7 | 0.393 |

## Age-related change in the influence of choice features on response times

In the main text of the manuscript, we reported results from a mixed-effects linear regression that demonstrated that the influence of the novelty and uncertainty of the selected option on participant response times varied across continuous age. Specifically, we observed that the slowing influence of novelty decreased with age while the slowing influence of uncertainty increased. To further elucidate these effects, we additionally ran separate mixed-effects linear regressions for each age group in which we modeled how choice features, age, and their interactions related to log-transformed response times. Here, we report the expected value, novelty, and uncertainty coefficients for each age group (*Appendix 1—table 5*).

**Appendix 1—table 5.** Model-derived effects (and standard errors) of expected value, novelty, and uncertainty on log-transformed reaction times across age groups.

| | Expected value coefficient | Novelty coefficient | Uncertainty coefficient |
|---|---|---|---|
| Children | −0.024 (0.01) | 0.074 (0.01) | 0.043 (0.01) |
| Adolescents | −0.023 (0.01) | 0.060 (0.01) | 0.068 (0.01) |
| Adults | −0.041 (0.01) | 0.053 (0.01) | 0.085 (0.01) |

## Accounting for quadratic age effects

In the main text of the manuscript, we used three age bins — children (*n* = 30, 8–12 years old), adolescents (*n* = 30, 13–17 years old), and adults (*n* = 62, 18–27 years old) — for data visualization and our computational modeling analyses. However, our adult age bin included both a wider range of ages and double the number of participants as our child and adolescent age bin. As such, we also visualized our results and ran our computational modeling analyses using four age bins, in which we split adults into young adults (*n* = 35, 18–22 years old), and adults (*n* = 27, 23–27 years old).

Our computational modeling results aligned with the findings we reported in the main text of the manuscript: Both young adults and adults were best fit by the familiarity-gated uncertainty model, according to protected exceedance probabilities (young adults: 0.988; adults: 0.993). Thus, these findings suggest that both young adult and adult participants considered option novelty when weighing reward uncertainty such that the aversive influence of uncertainty was attenuated for more novel options.

Our plots of choice data, however, suggested that younger adult participants demonstrated stronger aversion to uncertainty than older adult participants (*Appendix 1—figure 1*). Given the non-monotonic age-related increase in uncertainty aversion that these plots suggested, we re-ran our logistic regression analyzing how choice features influenced decision making across age, but with quadratic age as an additional fixed effect. We further included WASI scores as an interacting fixed effect in this model, because we observed group differences in reasoning ability (WASI score means (and standard deviations): Children: 115.4 (11.8); Adolescents: 111.4 (13.7); Young Adults: 113.6 (10.4); Adults: 107.7 (14.4)).

Here, we continued to observe robust effects of expected value, novelty, and uncertainty on choice behavior (*p*s < 0.001; *Appendix 1—table 6*). We also observed a linear age × uncertainty interaction effect, *p* = 0.027. We did not, however, observe a significant quadratic age × uncertainty interaction (*p* = 0.076). We did observe interactions between novelty and both linear and quadratic age (*p*s < 0.04), as well as interactions between novelty and WASI scores (*p* = 0.01).

**Appendix 1—table 6.** Influences on exploratory choice including quadratic age and WASI scores.

| | Odds ratio | 95% confidence interval | $X^2$ | *p* |
|---|---|---|---|---|
| Intercept | 1.00 | [0.95, 1.05] | | |
| Expected Value | 3.30 | [2.92, 3.74] | 168.0 | <0.001 |

*Appendix 1—table 6 Continued on next page*

*Appendix 1—table 6 Continued*

|  | Odds ratio | 95% confidence interval | $X^2$ | $p$ |
|---|---|---|---|---|
| Uncertainty | 0.87 | [0.82, 0.93] | 16.3 | <0.001 |
| Novelty | 1.36 | [1.30, 1.42] | 111.1 | <0.001 |
| Expected Value × Age | 1.82 | [0.76, 4.33] | 1.8 | 0.183 |
| Uncertainty × Age | 0.60 | [0.38, 0.94] | 4.9 | 0.027 |
| Novelty × Age | 0.71 | [0.51, 0.98] | 4.2 | 0.040 |
| Expected Value × Age$^2$ | 0.70 | [0.29, 1.68] | 0.6 | 0.429 |
| Uncertainty × Age$^2$ | 1.51 | [0.96, 2.37] | 3.2 | 0.076 |
| Novelty × Age$^2$ | 1.45 | [1.05, 2.01] | 4.9 | 0.026 |
| Expected Value × WASI | 1.33 | [1.18, 1.50] | 19.2 | <0.001 |
| Uncertainty × WASI | 0.96 | [0.89, 1.02] | 1.9 | 0.167 |
| Novelty × WASI | 1.06 | [1.01, 1.11] | 6.6 | 0.010 |
| EV × Age × WASI | 1.37 | [0.59, 3.17] | 0.5 | 0.463 |
| Uncertainty × Age × WASI | 0.71 | [0.46, 1.10] | 2.3 | 0.128 |
| Novelty × Age × WASI | 1.48 | [1.08, 2.02] | 5.9 | 0.016 |
| EV × Age$^2$ × WASI | 0.77 | [0.34, 1.74] | 0.4 | 0.534 |
| Uncertainty × Age$^2$ × WASI | 1.35 | [0.89, 2.06] | 2.0 | 0.162 |
| Novelty × Age$^2$ × WASI | 0.70 | [0.52, 0.94] | 5.2 | 0.022 |

## Age-related change in model-derived novelty and uncertainty biases

We further tested for age differences in the influence of novelty and uncertainty on choice behavior by examining parameter estimates from the reinforcement learning model that included separate novelty and uncertainty bias parameters. Though this model did not best fit the child, adolescent, or adult data, it well-captured qualitative patterns of choice behavior (see *Figure 5*) and its inclusion of separate novelty and uncertainty bias terms enables the examination of developmental differences in these choice features.

We ran linear regressions to examine how estimated parameters from the first-level, non-hierarchical model fits varied as a function of age. In line with the behavioral results reported in the main text, we found that the novelty bias did not significantly relate to age, *b* = 0.006, SE = 0.014, *p* = 0.699. However, the bias away from uncertainty increased with age, *b* = −0.014, SE = 0.006, *p* = 0.031. Together, these results further support our conclusion that age-related changes in exploration may arise from increasing aversion to reward uncertainty.

We further found that softmax inverse temperatures increased with age, *b* = 0.168, SE = 0.059, *p* = 0.005. This finding is in line with many prior developmental studies of reinforcement learning (*Nussenbaum and Hartley, 2019*), and indicates that older participants made choices that were increasingly driven by the choice utilities estimated by the model. We did not observe significant age differences in learning rates, *b* = −0.006, SE = 0.004, *p* = 0.104.

## Relation between reinforcement learning model parameters and subsequent memory

To examine the relation between reinforcement learning and subsequent memory for the high-value options, we ran linear regressions examining how overall memory test accuracy related to age, model parameters, and interactions between age and each model parameter. Because the best-fitting reinforcement learning model varied across age groups, we ran separate linear regressions for children (using parameters from the first-level fits of the novelty bias model) and adolescents and adults (using parameters from the first-level fits of the familiarity-gated uncertainty model). Across models, we did not observe any significant relations between reinforcement-learning model parameters and memory accuracy (*Appendix 1—table 7* and *Appendix 1—table 8*).

**Appendix 1—table 7.** Relation between parameter estimates from novelty bias model and memory accuracy in children.

| | Estimate | SE | t | p |
|---|---|---|---|---|
| Intercept | 0.25 | 0.03 | | |
| Age | −0.07 | 0.03 | −2.51 | 0.020 |
| Inverse Temperature | −0.03 | 0.04 | −0.71 | 0.484 |
| Learning Rate | −0.03 | 0.03 | −1.00 | 0.330 |
| Novelty Bias | −0.01 | 0.03 | −0.30 | 0.768 |
| Inverse Temperature × Age | 0.03 | 0.04 | 0.78 | 0.443 |
| Learning Rate × Age | 0.00 | 0.03 | 0.05 | 0.961 |
| Novelty Bias × Age | −0.01 | 0.04 | −0.27 | 0.793 |

**Appendix 1—table 8.** Relation between parameter estimates from familiarity-gated uncertainty model and memory accuracy in adolescents and adults.

| | Estimate | SE | t | p |
|---|---|---|---|---|
| Intercept | 0.25 | 0.01 | | |
| Age | −0.01 | 0.01 | −0.71 | 0.483 |
| Inverse Temperature | −0.01 | 0.01 | −0.76 | 0.450 |
| Learning Rate | −0.02 | 0.01 | −1.14 | 0.259 |
| Uncertainty Bias | −0.01 | 0.01 | 0.41 | 0.685 |
| Inverse Temperature × Age | 0.02 | 0.02 | 1.53 | 0.131 |
| Learning Rate × Age | 0.01 | 0.01 | −0.39 | 0.695 |
| Uncertainty Bias × Age | −0.01 | 0.01 | −0.70 | 0.489 |

## Reinforcement learning model validation: additional posterior predictive checks

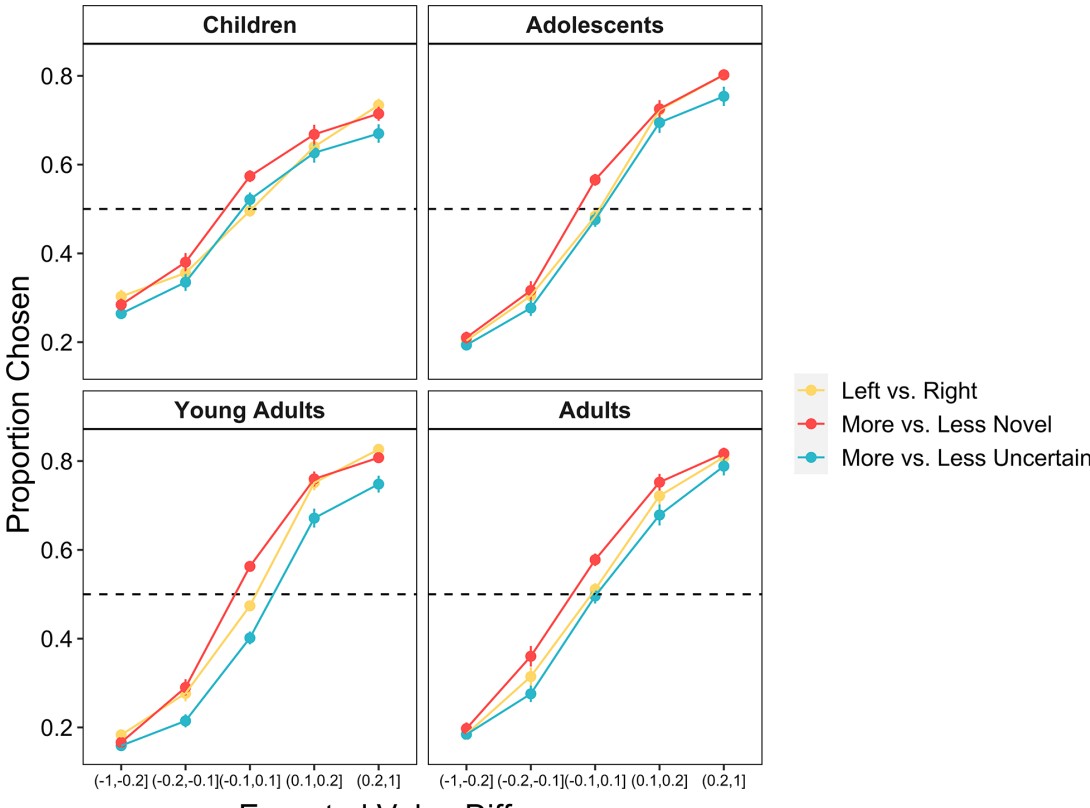

**Appendix 1—figure 1.** Influence of expected value, uncertainty, and novelty on choice behavior across age. The proportion of all trials in which participants (n = 122) chose the left, more novel, and more uncertain choice option as a function of the expected value difference between the options. Participants were more likely to choose options with greater expected value, higher novelty, and lower uncertainty ($ps$ < 0.001). The influence of novelty did not vary across linear and quadratic age, whereas uncertainty was more aversive in older participants ($p$ = 0.027). Uncertainty aversion did not vary as a function of quadratic age ($p$ = 0.076). Points indicate age group means and error bars show standard errors.

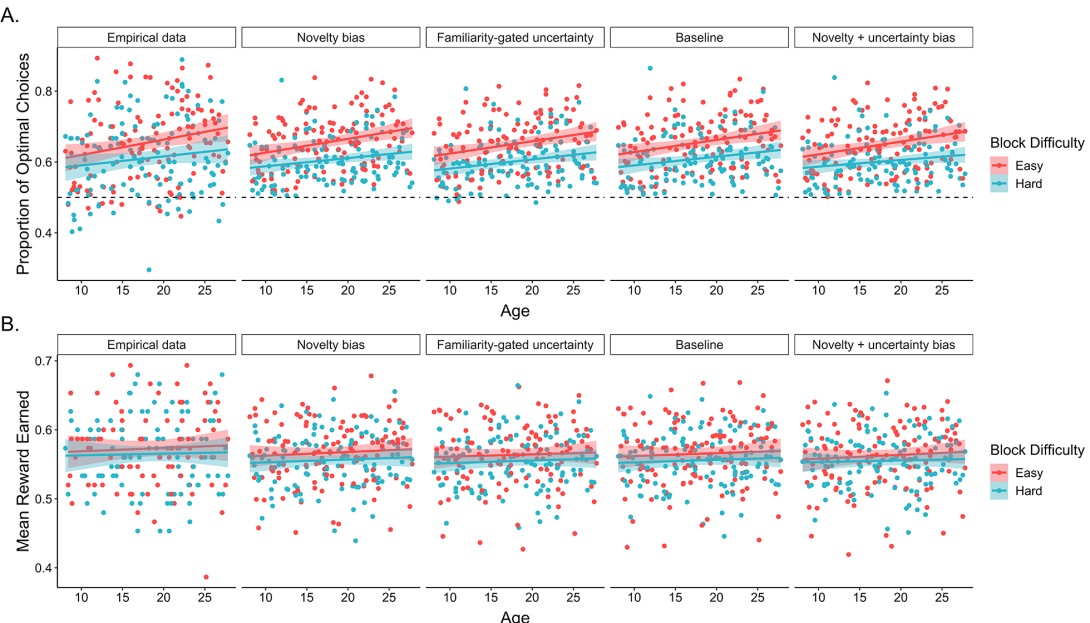

**Appendix 1—figure 2.** Model simulations of optimal choices and reward earned. The average proportion of trials in which both real and simulated participants (**A**) selected the optimal choice option and (**B**) earned reward. In both sets of plots, points represent participant averages in each block condition, lines show the best-fitting linear regression modeling the effect of age, and the shaded regions around them represent 95% confidence intervals. The dotted lines in panel A indicate chance-level performance.

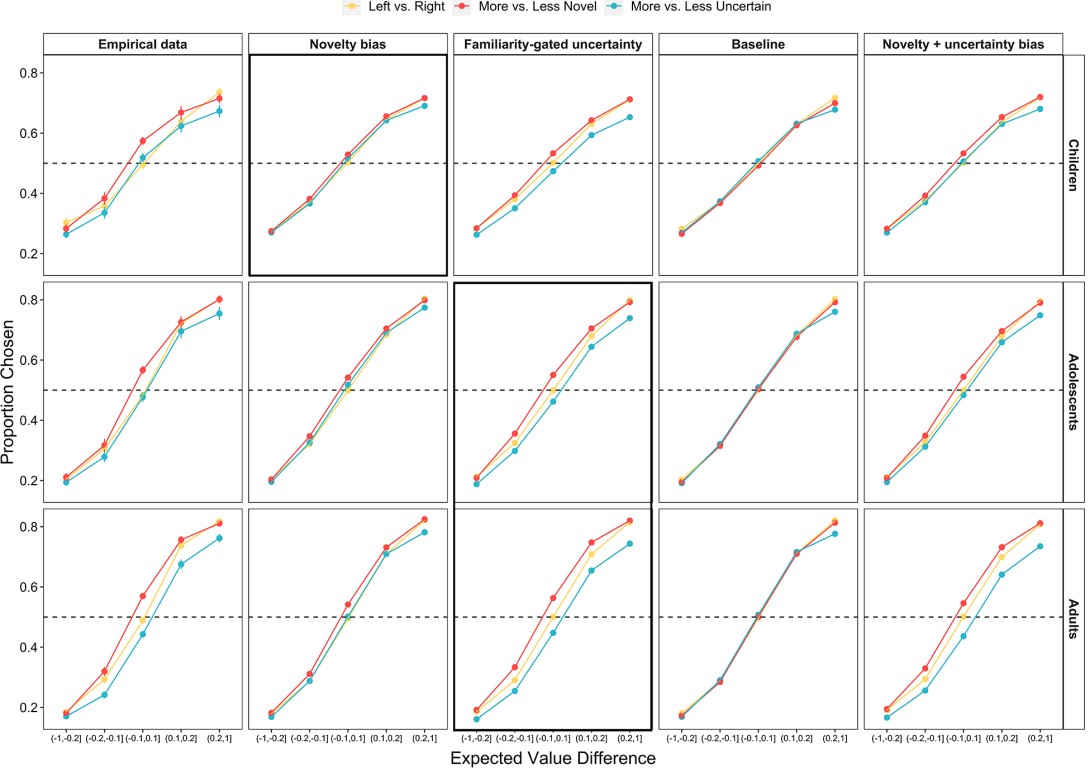

**Appendix 1—figure 3.** Influence of expected value, uncertainty, and novelty on choice behavior across age and model simulations. The proportion of all trials in which the real and simulated participants chose the left, more novel, and more uncertain choice option as a function of the expected value difference between the options. Points indicate age group means and error bars show standard errors. The thick black outlines indicate the best-fitting model for each age group.

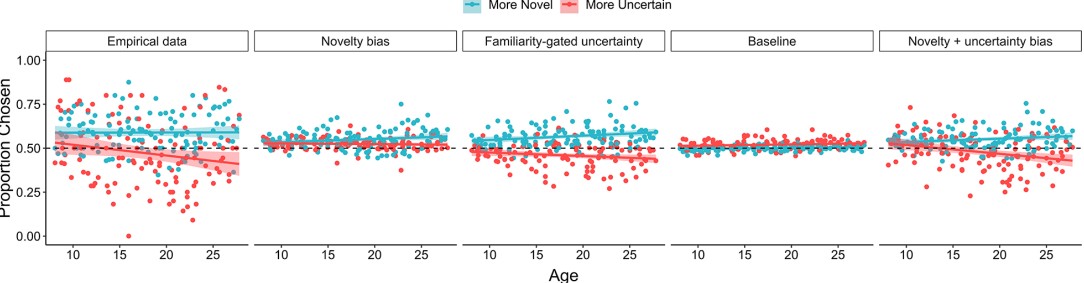

**Appendix 1—figure 4.** Influence of uncertainty, and novelty on choice behavior across age and model simulations for trials in which the choice options had similar-expected-values. The proportion of similar-expected-value trials (difference between the two options <0.05) in which participants chose the more novel and more uncertain option, plotted as a function of continuous age. The lines show the best-fitting linear regression lines and the shaded regions around them represent 95% confidence intervals.

