## [Editor Report]

This is an important study that investigates changes in novelty-seeking and uncertainty-directed exploration from childhood to adulthood. A wide age range of participants was tested using a well-suited task and performance was analyzed using a sophisticated model-based approach. The results provide compelling evidence that age-related changes in decision-making are driven by attenuation in uncertainty aversion in the presence of stable novelty-seeking from childhood to adulthood.

---

## [Decision Letter]

**Decision letter after peer review:**

Thank you for submitting your article "Novelty and uncertainty differentially drive exploration across development" for consideration by *eLife*. Your article has been reviewed by 2 peer reviewers, and the evaluation has been overseen by a Reviewing Editor, David Badre, and Timothy Behrens as the Senior Editor. The following individual involved in the review of your submission has agreed to reveal their identity: Charley M. Wu (Reviewer #1).

Essential revisions:

Though both reviewers found this to be an important investigation, they identified some points that could be addressed in revision to strengthen the support for the conclusions. Their detailed comments below are clear, so I won't copy them here. But they focus on four issues that would be essential to revise.

1) Modeling – Both reviewers had comments about ways that the modeling could be improved. R1 provided suggestions on how you can clarify and elaborate on the performance of the models. R2 suggested some new modeling, such as incorporating RT, given the sensitivity of decision time.

2) Statistics – R1 had several comments regarding the strength of the statistical evidence that could be clarified.

3) Novelty measure – R1 also raised questions about the definition of novelty. This is important to address, or at least clarify how you are defining it, given the centrality of this construct for your conclusions.

4) Reward uncertainty in children – Related to R2's comments on modeling, R2 also raised comments about the strength of evidence for the conclusion that children do not use reward uncertainty when making their decision when one considers measures like response time. This also seems important to address given its relationship to your conclusions about changes in this factor over development.

*Reviewer #1 (Recommendations for the authors):*

Nussenbaum, Martin, and colleagues present experimental data and computational modeling results to disentangle the role of uncertainty-directed and novelty-seeking exploration across childhood, adolescence, and adulthood. Their task uses an adapted version of a previously published decision-making task (Cockburn et al., 2022), where they analyze age-related differences in choice behavior (quantified in terms of value, uncertainty, and novelty), reaction times, and with various computational models incorporating different biases.

Overall, I found this work to be clearly written and well-motivated. Some priority claims in the introduction seemed a bit unnecessary though since previous work has also disentangled novelty and uncertainty (Stojić et al., 2020), and two papers already cited in the manuscript have looked at how exploration changes from childhood to adulthood (Giron et al., 2022; Schulz et al., 2019). However, I think this paper is interesting enough and provides a sufficiently new perspective without needing to rely on claims about being first.

The goal of the paper is to disentangle age-related changes in uncertainty-directed and novelty-seeking exploration. I believe that the behavioral and model-based analyses provide some (but perhaps limited) support for the idea that novelty-seeking stays fixed over the lifespan, whereas people become more uncertainty-averse as they get older.

While I greatly appreciate the use of multiple analyses to support their arguments, I have some reservations about the conclusions reached by this paper. These reservations hinge on potential weaknesses in statistical analyses, limited application of the computational models, and potential issues with the quantification of novelty.

Statistics

While the statistics were generally pretty clear, there was heterogeneity across behavioral analyses that seemed to lack justification. Moving from the proportion of optimal choices to total reward earned, it's unclear if the same random effects structure (as in the logistic regression) is applied or if a non-mixed effects model is used instead. If there is a specific reason for changing the statistical framework, it would be helpful to provide some justification.

Additionally, some of the behavioral effects seem rather weak when looking at the plots (e.g., changes in performance over age and age-related differences in choosing more vs. less uncertain options). While likelihood ratio tests are used to show that a model with the predictor is better than one without, the coefficients are not reported, making it difficult to assess whether the effect size is meaningful. Looking at Figure 2B, it wasn't obvious if children performed reliably worse than adults, or if the task is relatively simple such that a ceiling effect is reached. Reporting the regression coefficients (or the Odds Ratio for the logistic regression) and using a consistent mixed-effects structure would greatly help with interpreting these results.

Model results

One of the strengths of the paper is in using multiple analyses (e.g., choices, RTs, and models) to probe the main question. However, I felt that the model results were a bit underwhelming. It's clear a lot of work has gone into them, but we are given a relatively sparse account of how well they perform. Showing age-related changes to the parameters governing novelty and uncertainty bias would be very helpful in supporting the behavioral analyses in justifying the current conclusions. Currently, it's not reported if children's novelty bias differs from adolescents or adults or if novelty bias changes as a function of age. Second, beyond the rather 1-dimensional comparison provided in Figure 5, would you expect the models to also reproduce key behavioral signatures (e.g., in Figure 2 and Figure 3?)? It's also fine if this is out of the current scope, but a demonstration that age-related changes in uncertainty bias but not novelty is necessary to reproduce key aspects of behavior would provide the strongest support for the current claims. Lastly, the recovery results suggest potential misspecification with the winning model also accounting for a lot of the variance from other models. It would be very helpful to also report the inversion matrix (Wilson and Collins, 2019).

Measuring novelty

I found the quantification of stimulus novelty to be sensible but somewhat limited. It captures a uni-dimensional account of how often one has interacted with a given stimulus. But it doesn't capture how accustomed we are to other stimuli and how genuinely novel new stimuli may be. For example, for a naïve learner in the task, any new locations may be equally novel. However, over the course of learning, one gains familiarity with existing stimuli, such that new locations might be more surprising or unexpected. This familiarity effect would predict that novel stimuli in early blocks should differ from later blocks.

Perhaps a more holistic account of stimulus novelty could use a Dirichlet distribution over the n different hiding locations, where the concentration parameter α_i is updated for each experience of stimuli i. This would capture how expectations over all stimuli can inform novelty judgements about a particular new stimulus. I should clarify that this is a rather speculative suggestion, and I'm definitely not going to insist that you adopt it. Perhaps you've already looked into the effect of novelty over blocks and have already falsified this prediction. However, I am personally curious about how this would influence other downstream analyses. And since the forgetting mechanisms of the Bayesian learner can be applied to uncertainty but not to the current implementation of novelty, swapping to a Dirichlet distribution would also let you apply an equivalent decay process to the concentration parameters.

References:

Barr, D. J., Levy, R., Scheepers, C., and Tily, H. J. (2013). Random effects structure for confirmatory hypothesis testing: Keep it maximal. Journal of Memory and Language, 68(3). https://doi.org/10.1016/j.jml.2012.11.001

Giron, A. P., Ciranka, S. K., Schulz, E., van den Bos, W., Ruggeri, A., Meder, B., and Wu, C. M. (2022). Developmental changes resemble stochastic optimization. https://doi.org/10.31234/osf.io/9f4k3

Schulz, E., Wu, C. M., Ruggeri, A., and Meder, B. (2019). Searching for Rewards Like a Child Means Less Generalization and More Directed Exploration. Psychological Science, 30(11), 1561-1572.

Stojić, H., Schulz, E., P Analytis, P., and Speekenbrink, M. (2020). It's new, but is it good? How generalization and uncertainty guide the exploration of novel options. Journal of Experimental Psychology. General, 149(10), 1878-1907.

Wilson, R. C., and Collins, A. G. (2019). Ten simple rules for the computational modeling of behavioral data. *eLife*, 8. https://doi.org/10.7554/*eLife*.49547

*Reviewer #2 (Recommendations for the authors):*

This study deploys a modified value-guided decision-making task (adopted from Cockburn et al., 2022) to examine factors contributing to the commonly observed developmental shift toward exploitation. The paradigm permits a dissociation between sensory stimulus novelty and reward uncertainty, allowing the authors to investigate the effects of both factors on choice behavior across a large population of participants ranging from eight to 27 years. The authors apply a combination of linear mixed-effect modeling and reinforcement learning to examine age-related differences in the influence of novelty and reward uncertainty on choices and reaction times (RTs). Both types of analyses yield the same age-related differences: the choice behavior of adults is best captured by both sensory novelty (novelty-seeking) and reward uncertainty (uncertainty aversion), whereas the choice behavior of children is best captured by novelty, not uncertainty. Nevertheless, children's RTs are partly influenced by reward uncertainty. The authors conclude that the developmental shift toward exploitation may reflect an age-related increase in the aversion to reward uncertainty (as opposed to alterations in novelty-seeking) and that children do not factor uncertainty into their decision-making.

Strengths

The study deployed a well-suited paradigm (Cockburn et al., 2022) to examine age-related changes in decision-making behavior. The experiment enables a clean dissociation between (sensory) stimulus novelty and reward uncertainty and is amenable to both statistical and computational modeling of participant behavior. Notably, the authors took great care in adapting the paradigm for children and ensuring that all participants understood instructions involving the experimental manipulation (e.g., that the reward uncertainty resets every block while some stimuli might occur more often).

Another strength of this article is its multi-methodical approach to examining the core hypothesis that age-related changes in exploratory behavior are driven by how novelty and reward uncertainty influence choice evaluation. The statistical analyses appear sound and replicate prior findings. Moreover, the statistical analyses comport the results from a formal comparison of different reinforcement learning models (each implementing a different hypothesis). Notably, the authors demonstrate the validity of the comparison by showcasing model recovery on synthetic data.

Weaknesses

The only critical weakness I could identify pertains to the conclusion that children did not use reward uncertainty "to guide their decisions." This argument rests on the observation that children's choices were unaffected by reward uncertainty. However, the authors observed that RTs were affected by reward uncertainty, leading them to conclude that children may be aware of the reward uncertainty but don't factor it into the decision-making process. If RTs were regarded as a more sensitive measure than choices, then reward uncertainty might arguably still affect the decision process. This hypothesis may be better tested by fitting a drift-diffusion model to both choices and RTs and examining the effect of reward uncertainty on the rate of evidence accumulation. The advantages of this method would be that the model is fit using more information (choices and RTs) and that it allows dissociating mechanisms for biasing the decision process (e.g., rate of evidence accumulation) from mechanisms for delaying the decision process (response threshold).

Impact

From a conceptual point of view, the study contributes novel insights into developmental changes in decision-making behavior while, at the same time, offering a valuable replication of previous findings (Cockburn et al., 2022).

From a technical point of view, the adopted paradigm and modeling effort provide a valuable tool for other researchers to examine the distinctive contributions of novelty and reward uncertainty to human decision-making. Therefore, I applaud the authors for making their data, analysis, and code available on OSF.

1. Developmental shift in aversion to uncertainty. Is it the case that adults become more uncertainty-averse or that children are more uncertainty-seeking to begin with or both? Figure 3B suggests the former, although Figure 5 suggests the latter so it might be worth further (statistical) examination.

2. Effects of uncertainty on choices versus RTs in children. The manuscript argues that children "were able to track uncertainty, despite not using it to guide their decisions." This argument seems to rest on the observation that uncertainty did not affect choices but RTs. However, RTs are also impacted by the decision-making process and could be regarded as a more sensitive measure. Thus, I wonder if it is fair to argue that uncertainty did not influence children's choices. A cleaner examination of this question might involve fitting a drift-diffusion model (e.g., with the HDDM package) to both children's choices and RTs. An effect of uncertainty on the rate of evidence accumulation would indicate that uncertainty influences the decision-making process. In contrast, an effect on threshold would suggest that the decision is not biased based on uncertainty but generally slowed.

3. Perhaps I am missing something, but I wondered why the authors did not consider fitting a mixture RL model in which the interaction term (bias x novelty) was weighted depending on age (in addition to the main terms). The advantage of such an approach may be that, if hierarchically fitted, the weight could be directly examined.

---

## [Author Response]

Essential revisions:Reviewer #1 (Recommendations for the authors):Nussenbaum, Martin, and colleagues present experimental data and computational modeling results to disentangle the role of uncertainty-directed and novelty-seeking exploration across childhood, adolescence, and adulthood. Their task uses an adapted version of a previously published decision-making task (Cockburn et al., 2022), where they analyze age-related differences in choice behavior (quantified in terms of value, uncertainty, and novelty), reaction times, and with various computational models incorporating different biases.Overall, I found this work to be clearly written and well-motivated. Some priority claims in the introduction seemed a bit unnecessary though since previous work has also disentangled novelty and uncertainty (Stojić et al., 2020), and two papers already cited in the manuscript have looked at how exploration changes from childhood to adulthood (Giron et al., 2022; Schulz et al., 2019). However, I think this paper is interesting enough and provides a sufficiently new perspective without needing to rely on claims about being first.The goal of the paper is to disentangle age-related changes in uncertainty-directed and novelty-seeking exploration. I believe that the behavioral and model-based analyses provide some (but perhaps limited) support for the idea that novelty-seeking stays fixed over the lifespan, whereas people become more uncertainty-averse as they get older.While I greatly appreciate the use of multiple analyses to support their arguments, I have some reservations about the conclusions reached by this paper. These reservations hinge on potential weaknesses in statistical analyses, limited application of the computational models, and potential issues with the quantification of novelty.

Thank you for your positive evaluation and thoughtful comments on our manuscript. We have addressed all specific comments in turn, below.

In addition, we modified our introduction to remove our priority claim. Specifically, on p. 4, lines 74 – 82, we previously wrote:

“Though prior studies have found effects of novelty (Henderson and Moore, 1980; Mendel, 1965) and reward uncertainty (Blanco and Sloutsky, 2020; Meder et al., 2021; E. Schulz et al., 2019) on exploration and choice in early childhood, no prior studies have charted how their influence changes from childhood to early adulthood, leaving open the question of why children tend to explore more than adults.” We have now amended this text to read, “Though prior studies have found effects of novelty (Henderson and Moore, 1980; Mendel, 1965) and reward uncertainty (Blanco and Sloutsky, 2020; Meder et al., 2021; E. Schulz et al., 2019) on exploration and choice in early childhood, it is unclear how their relative influence changes from childhood to early adulthood, leaving open the question of why children tend to explore more than adults.”

StatisticsWhile the statistics were generally pretty clear, there was heterogeneity across behavioral analyses that seemed to lack justification. Moving from the proportion of optimal choices to total reward earned, it's unclear if the same random effects structure (as in the logistic regression) is applied or if a non-mixed effects model is used instead. If there is a specific reason for changing the statistical framework, it would be helpful to provide some justification.

Originally, we computed two measures of overall participant performance that were agnostic to exploration strategy: whether they selected the higher-valued stimulus on each trial (optimal choice) and the overall number of coins they found throughout the task (reward earned). For the first metric — optimal choice — we had one measure per participant *per trial*; thus, given multiple observations per participant, we analyzed this dependent variable with a logistic mixed-effects model. For the second metric — total reward earned — we had one measure per participant for the entire study. Thus, given a single observation per participant, we did not need to include random effects for each participant in the model, and analyzed this dependent variable with a linear regression.

That said, the reviewer’s comment made us realize there was no reason to treat these two variables differently (trial-wise for optimal choices and aggregated for total reward earned). Thus, we have now replaced our original ‘total reward earned’ linear regression with a new logistic mixed-effects model examining *trial-wise* reward earned for each participant. In addition, in response to a comment from Reviewer 2, to examine performance over the entire task, we have now modified both of these models to additionally include block number as an interacting fixed effect.

We have now edited this section of the manuscript on p. 6, lines 132 – 153, accordingly, writing:

“First, we examined whether participants learned to select the better options within each block of the task. On each trial, the optimal choice was defined as the option with the higher reward probability. A mixed-effects logistic regression examining the effects of within-block trial number, age, block difficulty, block number, and their interactions, with random participant intercepts and slopes across trial number, block difficulty, and block number revealed that participants learned to make more optimal choices over the course of each block, Odds Ratio (OR) = 1.11, 95% Confidence Interval = [1.07, 1.16], χ^2^(1) = 24.5, *p* <.001. In addition, participants learned faster, OR = 1.05 [1.01, 1.08], χ^2^(1) = 8.2, *p* = .004, and made more optimal choices in easy relative to hard blocks, OR = 1.11 [1.07, 1.15], χ^2^(1) = 25.5, *p* <.001 (Figure 2A). Performance also improved with increasing age, OR = 1.10 [1.03, 1.18], χ^2^(1) = 7.1, *p* = .008 (Figure 2A). While we did not observe a main effect of block number, we did observe a block number x block difficulty interaction, OR = .96 [.93,.99], χ^2^(1) = 7.3, *p* = .007, as well as a block difficulty x trial x block number interaction, OR = .94 [.91,.97], χ^2^(1) = 15.1, *p* <.001, such that performance differences between easy and hard blocks were greater earlier in the experiment. No other interactions reached significance (*p*s >.07).

We followed the same analysis approach to examine whether participants earned reward on each trial, though we removed the block difficulty and block number random slopes to allow the model to converge. Here, we similarly observed that performance improved across trials within each block, OR = 1.03 [1.00, 1.07], χ^2^(1) = 3.9, *p* = .049. Further, the rate at which participants learned to make rewarding choices was faster in easier versus harder blocks, OR = 1.03 [1.00, 1.06], χ^2^(1) = 3.9, *p* = .048, though the effect of block difficulty was greater in earlier blocks, OR = .97 [.94, 1.00], χ^2^(1) = 4.0, *p* = .045. No other main effects or interactions reached significance.”

We thank the reviewer for highlighting the needless inconsistency in our analysis approach and believe this updated analysis is a better examination of participant performance because it now accounts for trial-wise variance in choice behavior across both metrics.

Additionally, some of the behavioral effects seem rather weak when looking at the plots (e.g., changes in performance over age and age-related differences in choosing more vs. less uncertain options). While likelihood ratio tests are used to show that a model with the predictor is better than one without, the coefficients are not reported, making it difficult to assess whether the effect size is meaningful. Looking at Figure 2B, it wasn't obvious if children performed reliably worse than adults, or if the task is relatively simple such that a ceiling effect is reached. Reporting the regression coefficients (or the Odds Ratio for the logistic regression) and using a consistent mixed-effects structure would greatly help with interpreting these results.

First, we note that we followed a systematic approach when selecting the random-effects structure for our models, but did not describe this clearly in the original draft of our paper. We have now clarified this in the methods on p. 19, where we write:

“Models included random participant intercepts and slopes across all fixed effects, except where noted due to convergence failures.”

We agree with the reviewer that reporting effect sizes will help readers understand the strength of each reported effect. As such, we now report coefficients (and standard errors) for all linear models and odds ratios (and 95% confidence intervals) for all logistic regressions in the results. We thank the reviewer for this helpful suggestion.

As the reviewer notes, some of the behavioral effects reported in the manuscript (and shown in Figure 2) are small. In particular, overall performance on the task (as measured by trial-wise optimal choices and reward earned) does not show large developmental changes. When we redid our analysis of reward earned and examined trial-wise reward earned (instead of overall reward earned, as in the original submission), we did not observe significant age-related change. We have now amended the figure caption of Figure 2B to reflect the new analyses:

“Finally, we note that the primary purpose of this experiment was to examine age-related change in exploration strategy. Despite observing only small differences in overall performance across age, we see robust age differences in both value-guided (Age x Expected Value Odds Ratio = 1.22) and uncertainty-guided (Age x Uncertainty Odds Ratio = .89) decision making.”

Model resultsOne of the strengths of the paper is in using multiple analyses (e.g., choices, RTs, and models) to probe the main question. However, I felt that the model results were a bit underwhelming. It's clear a lot of work has gone into them, but we are given a relatively sparse account of how well they perform. Showing age-related changes to the parameters governing novelty and uncertainty bias would be very helpful in supporting the behavioral analyses in justifying the current conclusions. Currently, it's not reported if children's novelty bias differs from adolescents or adults or if novelty bias changes as a function of age.

Thank you for this suggestion. We agree that it would be highly informative to understand how the parameters that govern the novelty and uncertainty biases change with age. Originally, we did not conduct these analyses because participants of different ages were best fit by different models: Children were best fit by the novelty bias model whereas adolescents and adults were best fit by the familiarity-gated uncertainty model. Moreover, neither of these best-fitting models included both a novelty and an uncertainty bias parameter. The novelty bias model included only a novelty bias parameter and the familiarity-gated uncertainty model included only an uncertainty bias parameter. Thus, we could not use parameter estimates from either of these models to examine how the influence of both uncertainty and novelty on decision making changed with age.

The best model-based metrics of the influence of novelty and uncertainty on choice can be derived from the novelty + uncertainty bias model. This model includes a novelty bias parameter that determines the value at which novel stimuli are initialized, as well as an uncertainty bias parameter that determines how much each choice option’s uncertainty (the variance of recency-weighted experienced rewards) influences its overall utility. Because this model includes free parameters capturing our effects of interest that do not interact, we can more easily interpret age-related differences in parameter estimates from this model (as opposed to either of the two winning models). In addition, this model well-captured qualitative features of participant choices (see Figure 5).

Thus, to examine age-related change in model-estimated novelty and uncertainty biases, we analyzed parameter estimates from the novelty and uncertainty-biased model. In line with our behavioral results, we found that age was significantly related to the uncertainty model parameter, but not the novelty model parameter. Thus, across different analyses, we find convergent results.

We have now described our rationale for examining parameter estimates from this model, as well as our findings in Appendix 1 (pp. 9 – 10), where we write:

“We further tested for age differences in the influence of novelty and uncertainty on choice behavior by examining parameter estimates from the reinforcement learning model that included separate novelty and uncertainty bias parameters. Though this model did not best fit the child, adolescent, or adult data, it well-captured qualitative patterns of choice behavior (see Figure 5) and its inclusion of separate novelty and uncertainty bias terms enables the examination of developmental differences in these choice features.

We ran linear regressions to examine how estimated parameters from the first-level, non-hierarchical model fits varied as a function of age. In line with the behavioral results reported in the main text, we found that the novelty bias did not significantly relate to age, b = .006, SE = .014, *p* = .699. However, the bias away from uncertainty increased with age, b = -.014, SE = .006, *p* = .031. Together, these results further support our conclusion that age-related changes in exploration may arise from increasing aversion to reward uncertainty.

We further found that softmax inverse temperatures increased with age, b = .168, SE = .059, *p* = .005. This finding is in line with many prior developmental studies of reinforcement learning (Nussenbaum and Hartley, 2019), and indicates that older participants made choices that were increasingly driven by the choice utilities estimated by the model. We did not observe significant age differences in learning rates, b = -.006, SE = .004, *p* = .104.”

We refer to these new analyses on p. 11, where we write:

“Specifically, beyond the baseline model, we fit three additional models in which uncertainty and novelty exerted separable influences on choice behavior: a model augmented with a novelty bias that adjusted the initial hyperparameters of each option’s β distribution, a model augmented with an uncertainty bias that added or subtracted each option’s scaled uncertainty to its expected utility, and a model augmented with both biases. Corroborating our behavioral results, parameter estimates from the model with both a novelty and uncertainty bias revealed an age-consistent novelty preference but age-varying uncertainty aversion (see Appendix 1).”

Second, beyond the rather 1-dimensional comparison provided in Figure 5, would you expect the models to also reproduce key behavioral signatures (e.g., in Figure 2 and Figure 3?)? It's also fine if this is out of the current scope, but a demonstration that age-related changes in uncertainty bias but not novelty is necessary to reproduce key aspects of behavior would provide the strongest support for the current claims.

We thought the relative simplicity of our original Figure 5 would best help readers understand the behavioral predictions of the different models. Indeed the ‘key’ signature of the familiarity-gated uncertainty model is that it predicts divergent effects of novelty and uncertainty on choice.

That said, we appreciate the reviewer’s suggestion to further examine whether the model simulations also reproduce other signatures of participant behavior. We have now recreated Figure 2 and Figure 3 with data simulated from the novelty bias, familiarity-gated uncertainty, baseline, and novelty + uncertainty bias models. There are a few important insights that one can glean from these figures. First, all models make relatively similar predictions about participant performance (as indexed by optimal choices and reward earned). Second, the familiarity-gated uncertainty and the novelty + uncertainty bias models both predict divergent effects of novelty and uncertainty on choice behavior. Model comparison likely favors the familiarity-gated uncertainty model because it has one fewer parameter than the novelty + uncertainty bias model.

We have included all of these figures in Appendix 1. We reference them on pp. 12 – 13, lines 382 – 392, where we write:

“Model simulations revealed that the winning models well-captured qualitative features of behavioral choice data for each age group. For each model, we generated 50 simulated data sets using each of the 122 participants’ trial sequence and parameter estimates (for a total of 6,100 simulated participants per model). Data from these simulations demonstrated that the familiarity-gated uncertainty model generated the most strongly diverging effects of novelty and uncertainty on choice, in line with the adult and adolescent data (Figure 5; also see Appendix 1 – figures 2 – 4).”

Lastly, the recovery results suggest potential misspecification with the winning model also accounting for a lot of the variance from other models. It would be very helpful to also report the inversion matrix (Wilson and Collins, 2019).

Thank you for this suggestion. We have now modified Figure 6 to additionally include inversion matrices to examine the probability that data were generated by a specific simulated model given the recovered model.

We include the new figure in the methods on p. 22.

As both the confusion and inversion matrices show, data generated from the winning model for adolescents and adults — the familiarity-gated uncertainty model — cannot be distinguished from data generated from the familiarity-gated uncertainty model with an additional novelty bias. However, these results also demonstrate that models with familiarity-gated uncertainty *can* be distinguished clearly from models in which novelty and uncertainty do not interact. Thus, our model recoverability analyses indicate that we can be confident that adolescents and adults were influenced by interactive effects of novelty and uncertainty, but it is unclear whether they additionally demonstrated optimistic initialization of novel choice options.

We note this in the methods on p. 23, lines 736 – 748:

“Finally, the familiarity-gated uncertainty model with an additional novelty bias (implemented via asymmetric value initialization) was not recoverable at all and was almost always confused for the familiarity-gated uncertainty model. By attenuating the (generally aversive) influence of uncertainty on the utility of more novel stimuli, the familiarity-gated uncertainty model inherently implements a bias toward more novel options; thus, our task does not have the resolution to effectively determine whether there may also be an *additional* bias toward novelty instantiated via optimistic value initialization. However, the model recoverability results demonstrate that models that implement an interactive effect of novelty and uncertainty can be clearly distinguished from those that do not. Moreover, these model recoverability results indicate that the novelty bias model and familiarity-gated uncertainty model were highly distinguishable from one another, supporting our claim that children employed a value updating mechanism that was fundamentally distinct from the one used by adolescents and adults on our task.”

Measuring noveltyI found the quantification of stimulus novelty to be sensible but somewhat limited. It captures a uni-dimensional account of how often one has interacted with a given stimulus. But it doesn't capture how accustomed we are to other stimuli and how genuinely novel new stimuli may be. For example, for a naïve learner in the task, any new locations may be equally novel. However, over the course of learning, one gains familiarity with existing stimuli, such that new locations might be more surprising or unexpected. This familiarity effect would predict that novel stimuli in early blocks should differ from later blocks.Perhaps a more holistic account of stimulus novelty could use a Dirichlet distribution over the n different hiding locations, where the concentration parameter α_i is updated for each experience of stimuli i. This would capture how expectations over all stimuli can inform novelty judgements about a particular new stimulus. I should clarify that this is a rather speculative suggestion, and I'm definitely not going to insist that you adopt it. Perhaps you've already looked into the effect of novelty over blocks and have already falsified this prediction. However, I am personally curious about how this would influence other downstream analyses. And since the forgetting mechanisms of the Bayesian learner can be applied to uncertainty but not to the current implementation of novelty, swapping to a Dirichlet distribution would also let you apply an equivalent decay process to the concentration parameters.

This is an interesting prediction, though our current modeling approach should already capture these relative differences in novelty across blocks. Our regression models estimate the difference in novelty between two stimuli on every trial, where novelty is related to the number of times each stimulus has been encountered. As such, novelty differences increase over the course of the task as familiar stimuli become increasingly familiar.

Indeed, when we include block number as an interacting fixed effect in our mixed-effects linear regression examining the influence of novelty, uncertainty, and expected value on choice, our findings remain the same. We continue to observe main effects of value, novelty, and uncertainty, as well as interactions between age and both value and uncertainty on choice. Critically, we do not observe a significant block number x novelty interaction effect, X^2^(1) = 1.03, *p* = .310, nor do we observe any significant interactions with block number (all *p*s >.30). We have now included full results of this model in Appendix 1 on pp. 3 – 4.

Given the lack of block effects we observed in our regression, we decided not to model novelty with a Dirichlet distribution. Our current instantiation of the novelty bias, which sets the initial parameters of the β distribution over expected value for each choice option, already enables the modeling of ‘decaying’ effects of novelty. Because this β distribution is updated every time an option is selected, the influence of the initial novelty bias decays across trials, at a rate governed by each participant’s learning rate.

Reviewer #2 (Recommendations for the authors):1. Developmental shift in aversion to uncertainty. Is it the case that adults become more uncertainty-averse or that children are more uncertainty-seeking to begin with or both? Figure 3B suggests the former, although Figure 5 suggests the latter so it might be worth further (statistical) examination.

Thank you for raising this important point. We believe our data suggest that in this task, uncertainty aversion emerges and strengthens with increasing age. After controlling for the effects of novelty, we find that children are neither uncertainty-seeking nor uncertainty averse.

Figure 5 shows that children selected more uncertain choice options on slightly more than half of the equal expected value trials. However, this effect likely emerges due to the correlation between novelty and uncertainty in our task. Even though we specifically designed our experiment to disentangle novelty and uncertainty, we still observe a weak correlation (*r* = .143) between these two choice features. Importantly, Figure 5 also shows that this pattern of choices is well-captured by the simple novelty bias model, which does not take option uncertainty into account at all, indicating that in this task, a small preference for uncertain options can emerge through novelty-seeking alone.

In addition, when we run a mixed-effects logistic regression examining the influence of expected value, novelty, and uncertainty on children’s choices, we do not observe a significant effect of uncertainty on choice, OR = 1.07 [.945, 1.21], χ^2^(1) = 1.17, *p* = .280. We do observe significant effects of both value (OR = 2.27 [1.78, 2.91], χ^2^(1) = 26.8, *p* <.001) and novelty (OR = 1.35 [1.24, 1.48], χ^2^(1) = 28.4, *p* <.001). These results indicate that after controlling for novelty and expected value, children do not demonstrate any significant effect of uncertainty on choice.

We appreciate the reviewer’s suggestion to provide greater statistical evidence to arbitrate between these two potential accounts of uncertainty in our task. We have now included results from the child-only regression model in the main text of our manuscript on p. 9, lines 260 – 267:

“We further examined whether age-related change increases in uncertainty aversion were due to an early preference to engage with uncertain options or early indifference to uncertainty. To test these possibilities, we ran an additional mixed-effects logistic regression examining how expected value, uncertainty, and novelty influenced the choices of children only. Results indicated that children’s choices were significantly influenced by both expected value (OR = 2.27 [1.78, 2.91], χ2(1) = 26.8, *p* <.001) and novelty (OR = 1.35 [1.24, 1.48], χ2(1) = 28.4, *p* <.001). However, there was not an effect of uncertainty on choice, OR = 1.07 [.945, 1.21], χ2(1) = 1.17, *p* = .280, indicating no significant evidence for uncertainty-seeking in children.”

Finally, we note that the developmental trajectory we observed here (uncertainty indifference to uncertainty aversion with increasing age) is likely specific to our task. In our discussion, we emphasize that this may be due to the relatively short ‘horizon’ over which participants could exploit learned reward probabilities. On p. 14, lines 442 – 456, we write:

“Whereas novelty-seeking did not exhibit age-related change, uncertainty aversion increased from childhood to early adulthood, potentially reflecting developmental improvement in the strategic modulation of information-seeking. Prior studies of decision making have found that individuals across age demonstrate uncertainty aversion in some environments (Camerer and Weber, 1992; Payzan-LeNestour et al., 2013; Rosenbaum and Hartley, 2019) and uncertainty-seeking in others (Blanchard and Gershman, 2018; Giron et al., 2022; E. Schulz et al., 2019). These seemingly discrepant patterns of behavior may be explained by differences in the utility of resolving uncertainty across contexts. In environments where learned information can be exploited to improve subsequent choices, resolving uncertainty has high utility (Rich and Gureckis, 2018; Wilson et al., 2014), whereas in choice contexts with short temporal horizons, there is little opportunity to use learned reward information to improve future decisions (Camerer and Weber, 1992; Levy et al., 2010). In our task, individuals had a relatively short horizon over which to exploit reward probabilities that themselves required multiple trials to learn — children’s reduced uncertainty aversion may have emerged from insensitivity to the limited utility of gaining additional information about the most uncertain choice options (Somerville et al., 2017).”

2. Effects of uncertainty on choices versus RTs in children. The manuscript argues that children "were able to track uncertainty, despite not using it to guide their decisions." This argument seems to rest on the observation that uncertainty did not affect choices but RTs. However, RTs are also impacted by the decision-making process and could be regarded as a more sensitive measure. Thus, I wonder if it is fair to argue that uncertainty did not influence children's choices. A cleaner examination of this question might involve fitting a drift-diffusion model (e.g., with the HDDM package) to both children's choices and RTs. An effect of uncertainty on the rate of evidence accumulation would indicate that uncertainty influences the decision-making process. In contrast, an effect on threshold would suggest that the decision is not biased based on uncertainty but generally slowed.

Thank you for raising this important point. We considered analyzing our data with drift-diffusion models to jointly examine reaction times and choices in a single model. However, when applied to value-based decisions, the standard drift-diffusion model (DDM) assumes that the uncertainty of the value estimates for the two choice options on a given trial is equivalent (e.g., Fudenberg et al., 2018; Bakkour et al., 2019). The ‘right’ way to handle uncertainty in a DDM is an active empirical research question (e.g., Lee and Usher, 2023). Indeed, recent work has proposed multiple different ways in which uncertainty could be modeled. For example, drift rates may be influenced by the additive uncertainty of the two choice options, the difference in uncertainty between the high- and low-value option, or interactions between each option’s estimated value and estimated uncertainty, such that option values are scaled by option uncertainty (Lee and Usher, 2023). In addition, uncertainty may also influence decision thresholds in multiple different ways — thresholds may increase as additive uncertainty increases (Lee and Usher, 2023) or be influenced by the uncertainty difference between the two choice options. In addition to these possibilities discussed in the prior literature, it may also be the case that uncertainty’s influence on drift rates or decision thresholds interacts with novelty. And finally, the form of the influence of uncertainty on decision parameters may further vary with age.

Thus, we agree with the reviewer that fitting a DDM to examine the extent to which expected value, novelty, and uncertainty jointly influence responses and response times could address a very interesting question. However, we believe that effectively addressing this question is a big undertaking — requiring the careful fitting and validation of many different models for each age group — that is outside the scope of the current research. Effectively addressing how choice features influence rates of evidence accumulation and decision thresholds may also require conducting a new experiment in which participants are told to make each choice as quickly as possible. Here, participants were told that they had 4 seconds to make each decision and asked to respond within that timeframe; they were not told to make speeded responses.

The goal of our reaction time analyses was simply to determine whether children showed any sensitivity to the uncertainty of choice options. The fact that children were slower to select more uncertain choice options indicates that on some level, their behavior was influenced by option uncertainty. Thus, they demonstrate some degree of ‘uncertainty-tracking’ throughout the task, such that their lack of uncertainty aversion cannot be explained by complete insensitivity to option uncertainty.

We fully agree with the reviewer that the phrasing we used to interpret our results is misleading. Indeed, it is likely *not* fair to argue that uncertainty did not influence children’s choices — as the reviewer notes, RTs may reveal latent processes underlying choice behavior, and the fact that we see effects of uncertainty on children’s RTs indicates that uncertainty did influence children’s choices, even if it did not ultimately cause them to avoid more uncertain options. We have now carefully edited the discussion of our manuscript to better describe our results. We list those changes here for reference:

P. 13, lines 421 – 425:

Original text:

“While participants across age demonstrated a similar bias toward selecting more novel choice options, only older participants showed aversion to those with greater uncertainty. These findings suggest that children’s bias toward exploration over exploitation may arise from early indifference to reward uncertainty rather than heightened sensitivity to novelty.”

Revised text:

“While participants across age demonstrated a similar bias toward selecting more novel choice options, only older participants showed aversion to selecting those with greater uncertainty. These findings suggest that children’s bias toward exploration over exploitation may arise from attenuated aversion to selecting more uncertain options rather than heightened sensitivity to novelty.”

P. 14, lines 457 – 459:

Original text:

“Importantly, children’s slower response times when engaging with more uncertain options suggests that they were able to track uncertainty, despite not using it to guide their decisions.”

Revised text:

“Importantly, even though children’s choices were not uncertainty averse, children’s slower response times when engaging with more uncertain options suggests that they were able to track uncertainty.”

P. 15, lines 502 – 505:

Original text:

“Here, we demonstrated that the developmental shift from more exploratory to more exploitative behavior may arise from strengthening aversion reward uncertainty with increasing age, rather than from heightened sensitivity to novelty.”

Revised text:

“Here, we demonstrated that the developmental shift from more exploratory to more exploitative behavior may arise from strengthening aversion to selecting more uncertain options with increasing age, rather than from changes in novelty-seeking.”

In addition, we have added a section to our discussion in which we highlight the potential utility of DDMs in clarifying the precise way in which uncertainty may influence value-based decision making across age. On p. 15, lines 472 – 482, we write:

“Our observation of an influence of uncertainty on children’s reaction times suggests that uncertainty did indeed affect how children made value-based decisions. Future work could fit cognitive models to both participants’ choices and response times to investigate how, across age, uncertainty influences component cognitive processes involved in value-based decision-making. For example, researchers could use sequential sampling models to test different hypotheses about how value uncertainty — and its interactions with both expected value and novelty — influences both the rate at which participants accumulate evidence for a given option as well as the evidence threshold that must be reached for a response to be made (Lee and Usher, 2023; Wu et al., 2022). In addition, these approaches could be integrated with reinforcement learning models (Fonanesi et al., 2019) to gain further resolution into how the learned features of different options influence the choices participants make and the speed with which they make them.”

3. Perhaps I am missing something, but I wondered why the authors did not consider fitting a mixture RL model in which the interaction term (bias x novelty) was weighted depending on age (in addition to the main terms). The advantage of such an approach may be that, if hierarchically fitted, the weight could be directly examined.

Thank you for this suggestion. Indeed, it would be a more elegant approach to directly build age into our hierarchical model-fitting procedure, such that individual-level parameters are drawn from age-varying prior distributions (e.g., Xia et al., 2021). That said, this would be particularly valuable for making inferences about age-related change in parameter estimates, which was not a central focus of our analysis (though now included in Appendix 1 of the revised manuscript, see our response to comment 3 from reviewer 1).

Instead, we used model comparison to understand age differences in the algorithms underlying exploratory decision-making. For model-fitting and comparison, we used the cbm package (Piray et al., 2019), which offers one main advantage over other hierarchical model-fitting toolboxes: The extent to which the parameter estimates for a given participant influence the group-level prior is determined by how well the model captures their choice data. We note this advantage on p. 21, lines 714 – 717. We thought this modeling approach was particularly advantageous given that we wanted to make inferences about how well different models captured participant choices across age, while accounting for heterogeneity in model fits within each age group.

Unfortunately, incorporating age-varying priors into this model-fitting procedure is non-trivial. Indeed, we have corresponded with the package developer (Payam Piray) to inquire about the changes needed to implement group-level priors that vary based on age (or any participant-specific variable) and he confirmed that doing so is theoretically plausible but not currently possible within the current framework. Thus, given that parameter estimation was not a central focus of our analyses, we did not pursue this approach.

Finally, in response to reviewer 1, we have now included supplemental analyses examining age-related change in parameter estimates. In line with our existing behavioral analyses, we found an age-related increase in uncertainty aversion and no significant effect of age on the novelty bias parameter. We have described these new analyses in Appendix 1. Thus, these results suggest that our observed age effects are robust to diverse analytic approaches.